# Analysis of metagenome-assembled viral genomes from the human gut reveals diverse putative CrAss-like phages with unique genomic features

Natalya Yutin[1], Sean Benler[1], Sergei A. Shmakov[1], Yuri I. Wolf [1], Igor Tolstoy[1], Mike Rayko [2], Dmitry Antipov[2], Pavel A. Pevzner [3] & Eugene V. Koonin [1✉]

CrAssphage is the most abundant human-associated virus and the founding member of a large group of bacteriophages, discovered in animal-associated and environmental metagenomes, that infect bacteria of the phylum Bacteroidetes. We analyze 4907 Circular Metagenome Assembled Genomes (cMAGs) of putative viruses from human gut microbiomes and identify nearly 600 genomes of crAss-like phages that account for nearly 87% of the DNA reads mapped to these cMAGs. Phylogenetic analysis of conserved genes demonstrates the monophyly of crAss-like phages, a putative virus order, and of 5 branches, potential families within that order, two of which have not been identified previously. The phage genomes in one of these families are almost twofold larger than the crAssphage genome (145-192 kilobases), with high density of self-splicing introns and inteins. Many crAss-like phages encode suppressor tRNAs that enable read-through of UGA or UAG stop-codons, mostly, in late phage genes. A distinct feature of the crAss-like phages is the recurrent switch of the phage DNA polymerase type between A and B families. Thus, comparative genomic analysis of the expanded assemblage of crAss-like phages reveals aspects of genome architecture and expression as well as phage biology that were not apparent from the previous work on phage genomics.

[1] National Center for Biotechnology Information, National Library of Medicine, Bethesda, MD, USA. [2] Center for Algorithmic Biotechnology, Institute for Translational Biomedicine, St. Petersburg State University, St. Petersburg, Russia. [3] Department of Computer Science and Engineering, University of California-San Diego, La Jolla, CA, USA. ✉email: koonin@ncbi.nlm.nih.gov

During the past few years, the advances of metagenomics have transformed the field of virology by dramatically expanding the virosphere[1]. The human gut virome is one of the most intensely studied viromes on Earth thanks to the obvious health relevance[2]. However, the majority of the sequences in the gut virome, i.e., the nucleotide sequences from the virus-like particle fraction, show no significant similarity to any sequences in the current databases and thus represent virus dark matter[2–5]. This dark matter can be expected to consist, primarily, of viruses that are, at best, distantly related to the known ones, such that their identification by detecting signature virus proteins requires special effort. The most notable case in point is the discovery of crAssphage (after Cross-Assembly), the most abundant human-associated virus. The genome of crAssphage is a double-stranded (ds) DNA molecule of approximately 97 kilobase (kb) that was assembled from contigs obtained from multiple human gut viromes and appears to be circular in sequence analysis (by analogy with other phages with pseudo-circular genomes, this is, probably, a terminally redundant linear genome)[6]. The crAssphage is represented in about half of human gut metagenomes and in some of these accounts for up to 90% of the sequencing reads in the virus-like particle fraction. CrAssphage genome sequences have been detected in human gut metagenomes from diverse geographic locations, showing that crAssphage is not only the most abundant virus in some human gut microbiomes but also is widely spread across human populations[2,6–10].

Initial analysis of the crAssphage genome identified few genes with detectable homologs and failed to establish any relationships with other phages. However, subsequent searches of extended sequence databases using more powerful computational methods resulted in prediction of the functions of the majority of the crAssphage genes and delineation of an expansive group of crAss-like phages in diverse host-associated and environmental viromes[11]. The gene complements of the crAss-like phages show a number of distinct features, in particular, a predicted complex transcription machinery centered around an RNA polymerase (RNAP) with an unusual structure related to the structure of eukaryotic RNA-dependent RNAPs involved in RNA interference. Recently, the prediction of this distinct RNAP has been validated by structural and functional analysis demonstrating that this RNAP is a virion component that is translocated into the host cell together with the phage DNA and is responsible for the transcription of the phage early genes[12]. A phylogenomic analysis of 250 genomes of crAss-like phages led to the proposal of 4 distinct subfamilies and 10 genera[13].

Multiple lines of evidence indicate that the hosts of most if not all crAss-like phages belong to the bacterial phylum Bacteroidetes, which represent a dominant component of the human gut microbiome but are, largely, recalcitrant to growth in culture[6,11,14]. This host range accounts for the fact that the most abundant virus in the human virome remained unknown until the advent of advanced metagenomics and had been known only as a genome sequence for another 3 years. Nevertheless, recent concerted effort culminated in successful isolation of a crAss-like phage in a culture of *Bacteroides intestinalis*[15] and several phages of *Bacteroides thetaiotaomicron*[16].

Here we report an extensive analysis of large circular contigs from human gut metagenomes that resulted in the identification of nearly 600 diverse genomes of crAss-like phages, which are expected to comprise an order of viruses within the class Caudoviricetes, with 5 or 6 constituent families. Some groups of crAss-like phages show unusual genomic features, including frequent exchange of DNA polymerases between A and B families, extensive utilization of alternative genetic codes in late genes, and high density of group I self-splicing introns and inteins.

## Results

### Identification of crAss-like phages in human gut metagenomes and viromes

In order to obtain a set of complete virus genomes, we extracted all circular contigs (with exact identical regions 50–200 basepairs (bp) in length at the ends) from 5742 human gut metagenome and enriched virome assemblies resulting in 95,663 contigs (circular metagenome assembled genomes, hereafter cMAGs). Circular assemblies are not normally obtained with chromosomal sequences, the implication being that the set of cMAGs presumably contains a high fraction of plasmid and virus genomes.

To identify virus cMAGs, we searched the sequences with Hidden Markov Models (HMM) derived from 421 previously constructed alignments of conserved virus proteins (see "Methods"). Predicted proteins homologous to one or more of these conserved virus proteins were detected in 4907 cMAGs. This set of putative virus genomes was then searched with the profile for the large terminase subunit (TerL) of the crAss-like phage group, the protein with the highest sequence conservation. For the detected TerL homologs, phylogenetic analysis revealed a major, strongly supported clade that included 596 crAss-like phage cMAGs, which formed 221 clusters of distinct genomes sharing <90% of similarity at the DNA level (see "Methods" for details); these clusters are intended as a proxy for virus "species" (Supplementary Data 1). Together with the previously identified related viruses, these genomes comprise the "extended assemblage of crAss-like phages" (Fig. 1 and Supplementary Data 1). The emerging phylogenetic structure within this assemblage is generally compatible with and extends the findings of previous phylogenomic analyses[10,11,13].

We further sought to characterize the host range of crAss-like phages and, to that end, performed a search for CRISPR spacers matching the genome of these phages[17] (see "Methods" for details). This approach identified potential hosts for 466 of the 673 crAss-like phages. An overwhelming majority of the spacers with reliable matches came from CRISPR arrays in the genomes of different subdivisions of the phylum Bacteroidetes (Supplementary Data 1), supporting and expanding the previous host assignments. Nevertheless, several reliable spacer matches were detected in CRISPR arrays from bacterial phyla other than Bacteroidetes (gut Firmicutes and Proteobacteria, see Supplementary Data 1).

Another line of evidence on potential host range of crAss-like phages comes from BLAST searches of proteins encoded by crAss-like phages against a database of completely sequenced prokaryotic genomes (we limited the search to the set of complete genomes in order to avoid unreliable taxonomic assignments). Close matches (sequence identity ≥50% over at least 66% of length) were produced by this search suggesting relatively recent gene exchange between phages and their hosts. Among the 516 such matches, 325 (63%) came from Bacteroidetes, exceeding the random expectation about 11-fold (Supplementary Data 1). Arguably, some of these matches might originate from crAss-like prophages integrated in the genomes of Bacteroidetes, which, again, would support the host assignment.

### Conserved and group-specific genomic features in the extended crAss-like phage assemblage

Examination of the TerL tree suggests a coarse-grain structure of the relationships among the crAss-like phages, which include the human gut "crAss-virome," the previously analyzed environmental contigs and several independently isolated phages (ftp://ftp.ncbi.nih.gov/pub/yutinn/crassfamily_2020/all_genomes.fa and at https://doi.org/10.5281/zenodo.4437596). The monophyly of the previously proposed alpha-gamma, beta, and delta subfamilies of crAss-like phages

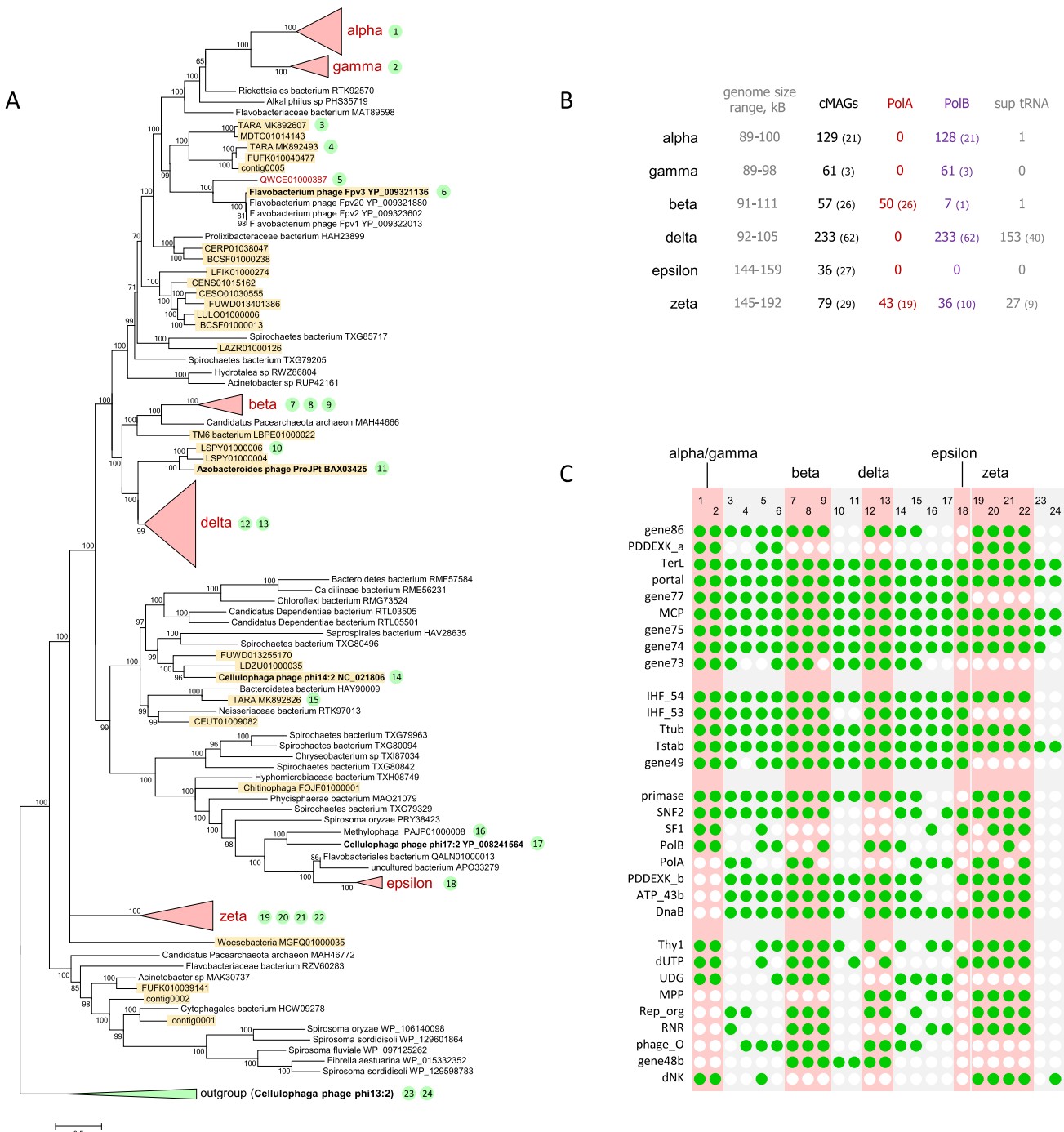

**Fig. 1 The crAss-like phage assemblage. a** Phylogenetic tree of the TerL protein of crAss-like phages. CrAss-like groups are marked with red color. Previously analyzed crAss-like sequences are highlighted in yellow. Cultured phages are marked with bold font. Numbers in green circles indicate genomes for which the gene conservation pattern is shown in **c**. **b** Some key genome features of CrAss-like cMAGs. In parentheses are numbers on distinct genomes (sharing <90% of similarity at the DNA level). **c** Pattern of gene conservation in crAss-like phages. TerL terminase large subunit, portal portal protein, gene77 putative structural protein (gene 77), MCP major capsid protein, gene75 putative structural protein (gene 75), gene74 putative structural protein (gene 74), gene73 putative structural protein (gene 73), IHF_54 IHF subunit (gene 54), IHF_53 IHF subunit (gene 53), Ttub Tail tubular protein (P22 gp4-like), Tstab tail stabilization protein (P22 gp10-like), gene49 uncharacterized protein (gene 49), gene86 putative structural protein (gene 86), PDDEXK PD-(D/E)XK family nuclease, DnaB phage replicative helicase, DnaB family, primase DnaG family primase, SNF2 SNF2 helicase, SF1 SF1 helicase, ATP_43b AAA domain ATPase, PolB DNA polymerase family B, PolA DNA polymerase family A, Thy1 thymidylate synthase, dUTP dUTPase, UDG Uracil-DNA glycosylase, MPP metallophosphatase, Rep_Org replisome organizer protein, RNR ribonucleotide reductase, phage_O bacteriophage replication protein O (gene10 of IAS virus), gene48b phage endonuclease I, dNK deoxynucleotide monophosphate kinase.

was confirmed; in addition, two previously unknown groups were identified, dubbed here Zeta and Epsilon (Fig. 1 and Supplementary Data 1). The Zeta and Epsilon groups comprise the deepest branches in the TerL tree outgrouped by environmental phages that are not considered parts of the crAss-like assemblage. The 5 groups, Alpha-Gamma, Beta, Delta, Epsilon, and Zeta, account for 595 of the 596 cMAGs in the human gut crAss-virome. One cMAG did not belong to any of these 5 clades but rather grouped with the Flavobacterium phage Fpv3 (a phage infecting a fish pathogenic bacterium, RefSeq: NC_031904); this might not be a native human gut phage (Fig. 1a). The monophyly of each of the five groups was confirmed by construction of additional trees, reconstructed from MCP (major capsid protein), portal, and gene75 amino acid sequences (ftp://ftp.ncbi.nih.gov/pub/yutinn/crassfamily_2020/trees and at https://doi.org/10.5281/zenodo.4437596).

The Alpha-Gamma group is best characterized and annotated and includes the "crAssphage clade" (Alpha) where the original crAssphage belongs[10,11]. It is the most diverse group of crAss-like phages; the average distance from the group common ancestor to the leaves comprises 65% of the total depth of the tree (averaged across the TerL, MCP, portal, and gene75 trees; see sheet "depth" in Supplementary Data 1). In our previous analysis[11], the Beta group was represented by a single complete genome, immune deficit-associated phage; since then, this group has been expanded to include a variety of human gut metagenomic contigs[13] along with ΦCrAss001, the first cultured crAss-like phage[15], and two other recent isolates, DAC15 and DAC17[16]. The Delta group was initially represented by several incomplete phage genomes[11] and was subsequently expanded[13]. The Delta group is currently the largest group in the gut crAss-virome (Fig. 1b), although it is the most compact one, with the intra-group diversity spanning only 24% of the tree depth (Supplementary Data 1). The Epsilon and Zeta groups have not been recognized previously. The Epsilon group is dominated by gut cMAGs but additionally includes Cellulophaga phage phi17:2 and two GenBank contigs, Methylophaga PAJP01000008 and Flavobacteriales bacterium QALN01000013. The phage genomes in the Epsilon group are about 150 kb long, that is, roughly 50% larger than the original crAssphage genome. The Zeta group consists entirely of MAGs from the human gut microbiomes analyzed here and some

previously reported human gut and oral metagenome contigs[18], with few to no homologs, detectable by BLASTP search, among known phages or any sequences in GenBank. One exception is a syntenic block of five homologs of unknown function encoded by Zeta family members and phage DAC16, a Bacteroides-infecting phage outside of the crAss-like phage assemblage[16] (Supplementary Fig. 1). The Zeta phages show substantial within-group diversity (the second most diverse group spanning 59% of the tree depth, Supplementary Data 1) and have genomes in the range of 145–192 kb, some of which are nearly twice as large as the original crAss-like genomes.

The genomes of crAss-like phages contain three readily discernible blocks of genes that encode: (1) virion components and proteins involved in virion assembly, (2) components of the replication apparatus, and (3) components of the transcription machinery (Fig. 2). The structural block is overall the most highly conserved one and includes the eight genes that are shared by all crAss-like phages and encode MCP, TerL, portal protein, Integration Host Factor (IHF), tail stabilization protein, tail tubular protein, along with uncharacterized genes 74 (Beta and Delta groups encompass tandem duplications of this gene) and 75; in addition, gene 86 that is adjacent to terL encodes a small protein that is likely to represent the terminase small subunit (TerS) despite the lack of detectable similarity with TerS of other phages (the genes are numbered according to the crAssphage genome annotation). The presence of the IHF, which is likely to contribute to DNA compactification during virion assembly, and of the two conserved uncharacterized genes in the structural block can be considered the genomic signature of the crAss-like phages. Three more genes including a paralog of IHF are present in (nearly) all crAss-like genes except for the zeta group, and one more gene is missing in both the zeta and epsilon groups (Fig. 1c).

In the replication block, the four highly conserved genes are DnaG family primase, two helicases of DnaB and SNF2 families, and an AAA+ superfamily ATPase. Each of these genes is represented in the majority of crAss-like phages including the zeta group which implies that they were present in the last common ancestor of these phages but were differentially lost in the alpha-gamma, delta, or epsilon group (Fig. 1c and Supplementary Fig. 2). In accord with previous observations, phylogenetic analysis of the primase from the extended set of

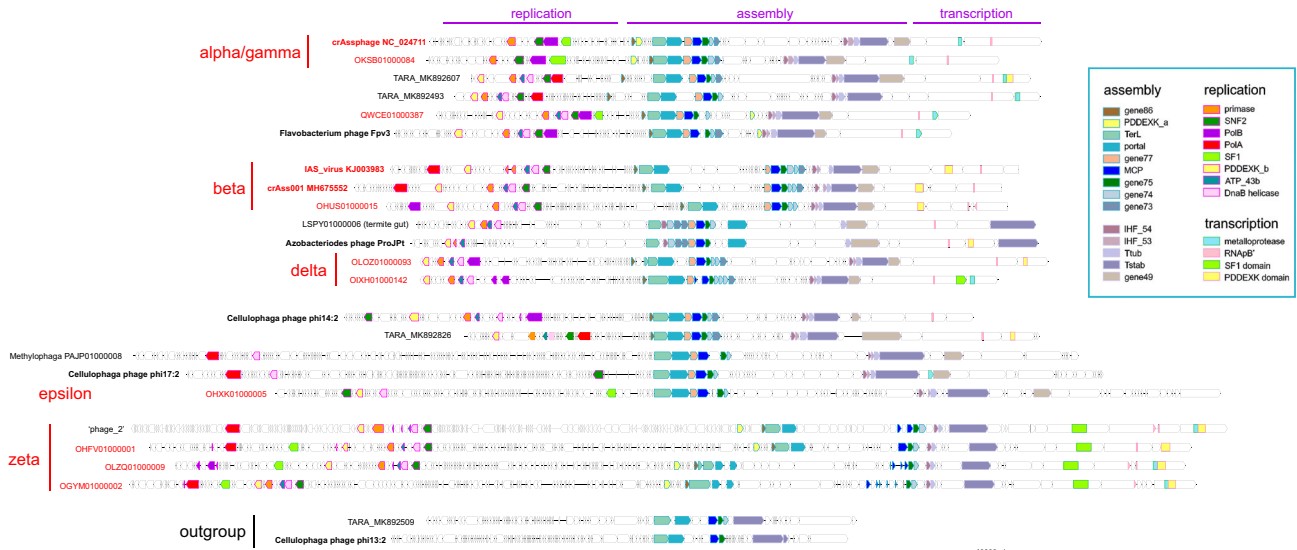

**Fig. 2 Genome organizations of representative crAss-like phages.** The circular genomes were linearized by breaking the circle after the transcription gene block. Conserved proteins are abbreviated as in Fig. 1c; RNApB' marks the conserved DxDxD motif of the large RNAP subunit homologous to the bacterial β' subunit.

crAss-like phages demonstrates the single origin of the phage primase from DnaG of Bacteroidetes (Supplementary Fig. 3). With the exception of the epsilon group and a few phages from environments other than human gut, the crAss-like phages encode a DNA polymerase (DNAP) of either A or B family (PolA and PolB, respectively). The evolution of the DNAPs of crAss-like phages is discussed in detail in the next section. In addition to the proteins directly involved in replication, crAss-like phages encode several enzymes of nucleotide metabolism, in particular, thymidylate synthase (Thy1), dUTPase, Uracil-DNA glycosylase (UDG), and ribonucleotide reductase (RNR). In this group of enzymes, Thy1, dUTPase and RNR appear to be ancestral among the crAss-like phages, albeit lost in some members (Fig. 1c). As this analysis shows, some of the crAss-like phages, particularly those in the Zeta group that have the largest genomes in the entire assemblage, encode a complex replication apparatus complemented by enzymes of deoxynucleotide metabolism. By contrast, phages in the Epsilon group encode a minimal set of replication machinery components.

The transcription gene block that is present in all crAss-like phage genomes, except for the Epsilon group, consists of several large, multidomain proteins, one of which contains the DxDxD motif that is conserved in one of the large subunits of bacterial, archaeal, and eukaryotic RNAPs and is an essential part of the catalytic site (Fig. 3). The predicted RNAPs of crAss-like phages are extremely divergent in sequence from all known RNAPs and from each other, suggestive of unusually high evolutionary rates. The recent structural and functional study of the RNAP of *C. baltica* phage phi 14:2 confirms the involvement of this large protein in the transcription of the phage early genes and shows that it contains two double-psi beta-barrel (DPBB) domains within a single polypeptide, unlike cellular RNAPs in which the two DPBB domains belong to the two largest subunits[12].

A distinct feature of the crAss-like phages (with the exception of some in the Epsilon group) that, to our knowledge, has not been observed in other phages is the presence of at least one nuclease of the PDDEXK family, whereas many crAss-like phages encode 2 or even 3 such nucleases (Fig. 2). Apparently, these nucleases perform more than one function in phage reproduction because one variety is encoded within the structural block (PDDEXK_a) and another one within the replication block (PDDEXK_b), whereas some phages contain a PDDEXK domain in one of the large proteins associated with transcription (Figs. 2 and 3).

The genomes of many crAss-like phages encompass blocks of short genes that have no detectable homologs outside the crAss-like assemblage (Fig. 2). Given that, as indicated above, many crAss-like phages are targeted by CRISPR-Cas systems, we suspected that at least some of these genes could encode anti-CRISPR proteins (Acr). Most of the Acrs are small proteins that are encoded in arrays of short genes in the genomes of a variety of bacterial and archaeal viruses and lack detectable homologs except in closely related viruses[19,20]. Using the recently developed machine-learning method for Acr prediction[21], we searched the crAss-like phage genomes but failed to confidently predict any Acrs. We nevertheless suspect that some if not most of these proteins target host defense systems including some putative Acrs with features distinct from those previously identified.

Gene complements of crAss-like phages are highly diverse both within and across groups such that about 20% of the genes in each phage genome have no detectable homologs in other members of the crAss-like assemblage. Rarefaction analysis indicates that the currently sampled crAss-like pangenome is far from saturation (Fig. 4), so that numerous unique genes are expected to appear in additional crAss-like phage genomes.

**CrAss-like phages dominate the human gut virome**. Among the sequence reads mapped to putative virus genomes (see "Methods" for details), crAss-like phages accounted for 86.7% (Fig. 5 and Supplementary Data 2), the next most abundant group being Tevenvirinae (3.8% of reads). Among the crAss-like phages, the Alpha-Gamma group accounts for more than half of the reads (53.7%). The Delta, Epsilon, and Zeta groups are almost equally abundant, each comprising 13–18% of the reads, whereas the Beta group is markedly less common (1.6%). Two species-level clusters of cMAGs, Alpha_1_1 (including the crAssphage) and Zeta_8_2 (including cMAGs OGOT01000012 and OJOE01000015), comprise over half of all crAss-like phage-specific reads (46.4 and 8.0%, respectively).

**Switching of DNA polymerases in crAss-like phages**. With the exception of the epsilon group, crAss-like phages contain a gene within the replication block that encodes a DNA polymerase of either A family or B family (the two families contain distantly related variants of the core Palm domain[22]; Fig. 1b). In some of the groups of crAss-like phages, one or the other DNA polymerase is used exclusively or preferentially, for example, PolB in the

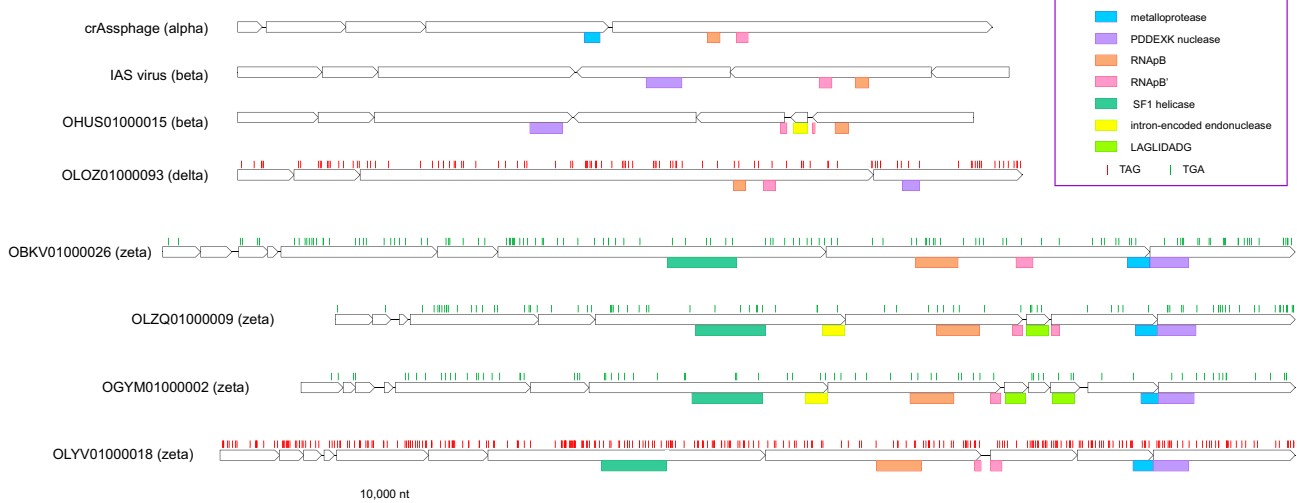

**Fig. 3 Transcription gene block in crAss-like phages.** The predicted genes are shown by empty block arrows, and recognized conserved domains are indicated by colored rectangles. Vertical bars indicate in-frame stop codons. LAGLIDADG is intron-encoded endonuclease maturase.

Alpha-Gamma group and PolA in the beta group, but in other groups, in particular, Zeta, PolA and PolB are mixed on a much finer scale (Fig. 6 and Supplementary Fig. 4). The location of the

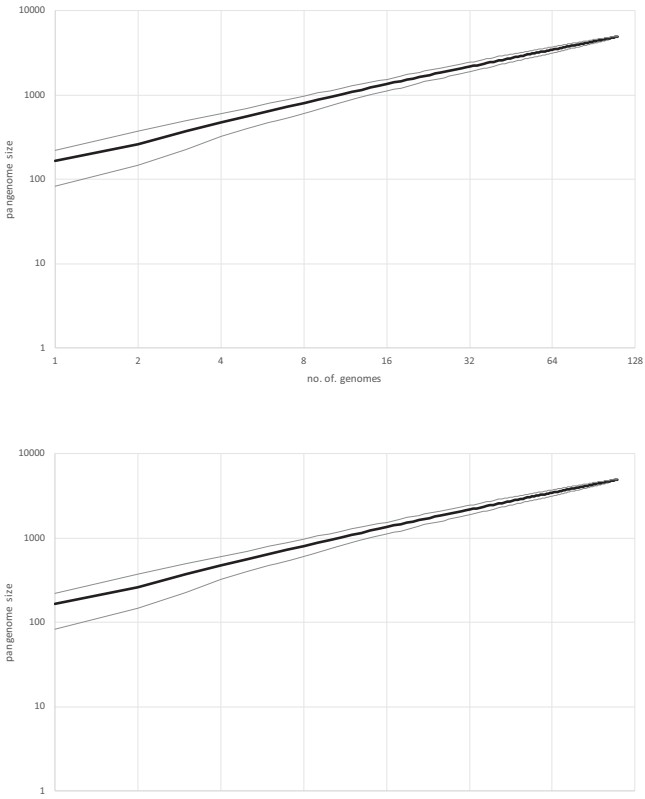

**Fig. 4 Rarefaction analysis of the crAss-like phage pangenome.** The results of 1001 rarefaction runs for 110 representative genomes are shown. The black line shows the median, and the gray lines show the 5 and 95 percentiles.

DNAP gene is mostly consistent within a group but can differ between the groups, for example, upstream of the primase gene in Alpha and downstream of the primase in Beta (Supplementary Fig. 2). Notably, in the groups where the two DNAP varieties are present alternatively, they typically occupy the same position, that is, appear to be replaced in situ (Supplementary Fig. 5).

Because none of the extant crAss-like phages encode two DNAPs, it is natural to assume that their last common ancestor also possessed only one DNAP. The extant distribution of PolA and PolB, together with the phylogenetic analysis of both DNAP families, imply numerous, independent events of both xenologous gene displacement (by a DNAP gene of same family but from a distant clade) and non-orthologous gene displacement (PolA with PolB or vice versa), often occurring in situ, that is, without disruption of the gene organization in the phage genomes. Thus parsimonious reconstruction of the ancestral state does not seem to be feasible, so that both PolA and PolB are equally suitable candidates for the role of the ancestral DNAP. Phylogenetic analysis of both PolA and PolB indicate their monophyly among the crAss-like phages except, possibly, for some distant variants in the zeta group, so that displacement appears to have been involved, primarily, among the crAss-like phages (Supplementary Fig. 4). The presence of different DNAPs in equivalent positions in many pairs of closely related phages implies that DNAP gene displacement occurs by homologous recombination. Given that the five groups of crAss-like phages remain, to a large extent, monophyletic in the DNAP phylogenies (Supplementary Fig. 4), it seems likely that, within the groups, displacement occurs, primarily, by homologous recombination between closely related genomes during coinfection, followed by sequence divergence. However, the observed mixing between groups in the DNAP phylogenies, for example, the presence of multiple members of the zeta group within the beta branch of PolB, implies gene displacement between distantly related phages that require mechanisms other than homologous recombination (Supplementary Fig. 6).

**Alternative coding strategies in crAss-like phages.** Typically, annotation of a bacterial virus genome starts with predicting

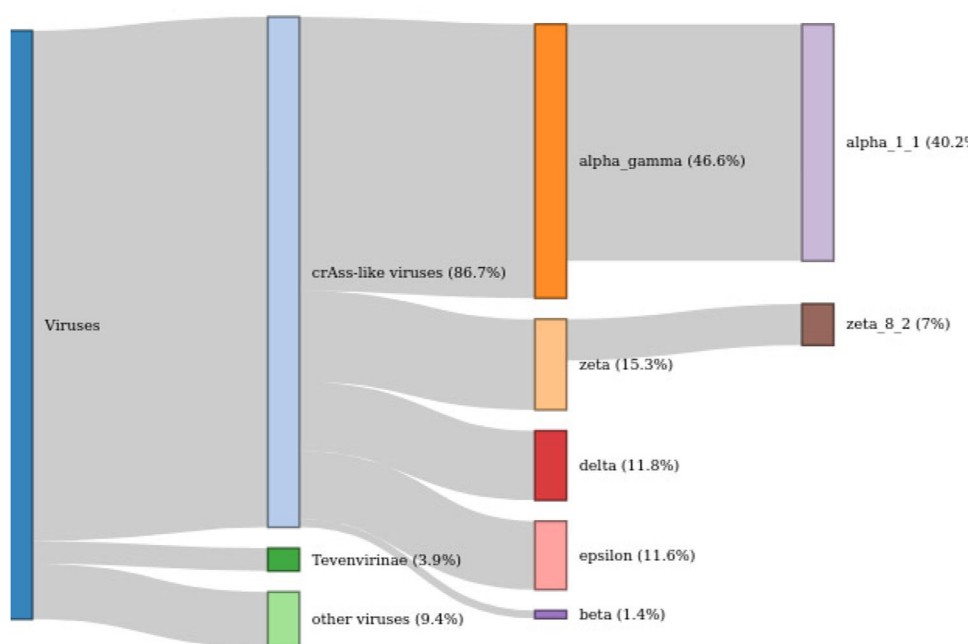

**Fig. 5 The crAss-like phages dominate the human gut virome.** The schematic shows the fractions of sequence reads mapped to different groups of virus genomes. The total represents 49,540,005 reads mapped to virus sequences.

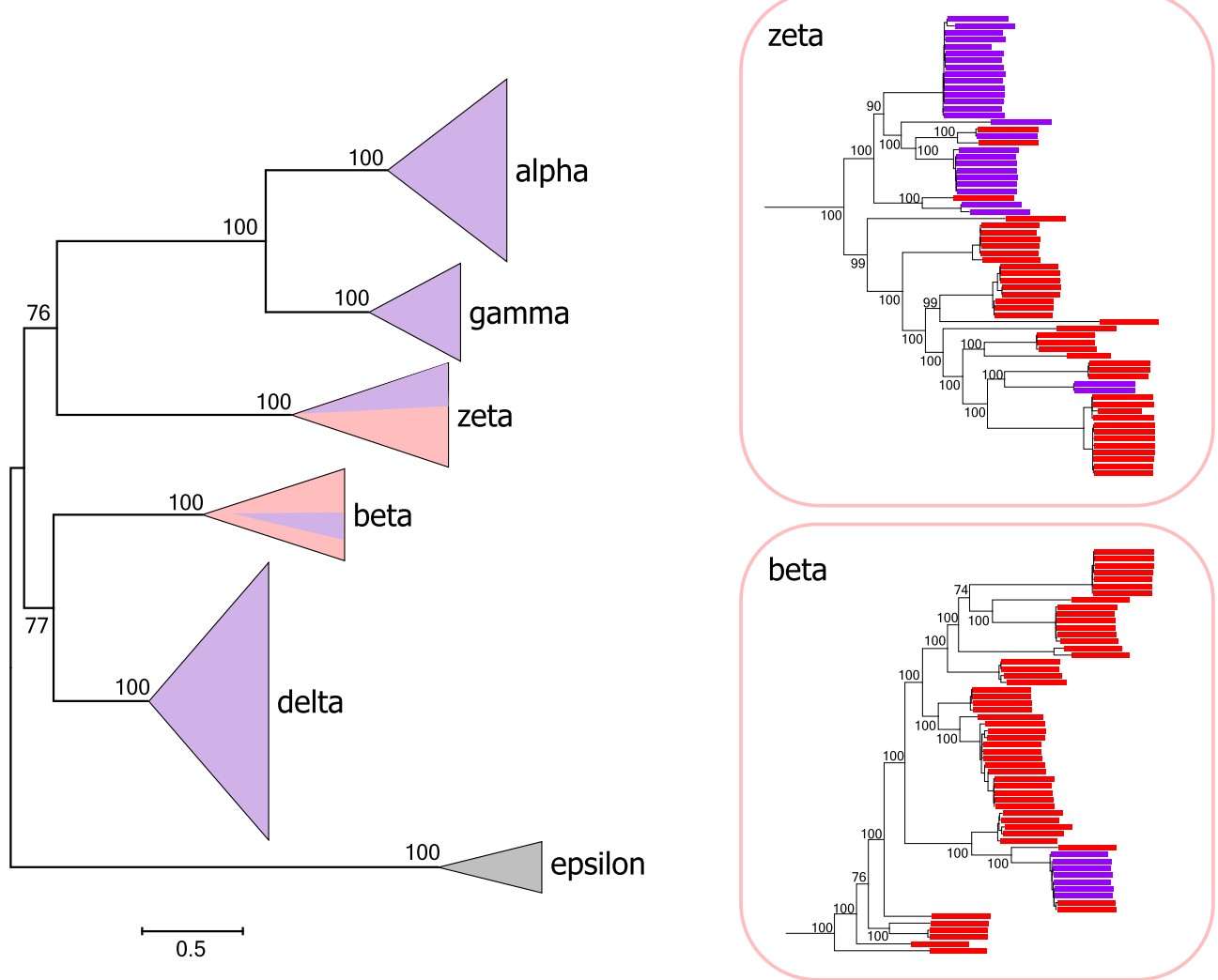

**Fig. 6 DNA polymerase switches in the evolution of crAss-like phages.** The left panel shows a schematic phylogenetic tree of TerL for the entire crAss-like phage assemblage, with the branches collapsed into triangles. The right panels show the expanded beta and zeta branches. The portions of the triangles (left panel) and the tree leaves (right panels) are colored by the DNAP type, red for PolA and purple for PolB.

protein-coding genes (open reading frames (ORFs)) and producing the conceptual translation of the ORFs into protein sequences[23]. When attempting such annotation of the genomes of the crAss-like assemblage, we encountered several phenomena that substantially impeded the analysis. Many of the genomes contain short ORFs encoding fragments of conserved proteins. Some of these fragments lie in the same frame and are interrupted only by standard stop codons; others are also in the same frame but are separated by longer in-frame inserts; yet others are in different frames and are separated by nucleotide sequences that are rich in stop codons in all co-directed frames. Even more unusually, some of the crAss-like phage genomes, particularly, those of the Zeta group, contain regions that do not encode any recognizable proteins but rather are occupied by short ORFs of different polarities. Many of the predicted protein sequences in these regions do not match even when the rest of the phage genomes are closely similar.

Some of the crAss-like phage genomes (for example, Eld241-t0_s_1 of the delta group; Supplementary Fig. 7) contain almost no long ORFs when translated under the standard genetic code with three standard stop codons. However, these genomes contain many ORFs encoding homologs of known phage proteins including TerL, portal, and RNAP subunits when translated with

an alternative code in which TAG encodes an amino acid (most likely, glutamine) instead of being a stop codon. Thus these phages apparently use such an alternative genetic code; indeed, to retrieve the characteristic set of the signature proteins of crAss-like phages, the entire genome of Eld241-t0_s_1 had to be translated using the code with TAG reassigned for glutamine[13].

Alternative genetic codes in phages including extensive opal (TGA) and amber (TAG) stop codon reassignments have been reported previously, especially, in human-associated metagenomes[18,24,25]. We used the presence of standard stop codons in several readily identifiable, nearly ubiquitous genes (TerL, MCP, portal) as evidence of likely stop codon reassignment, and the contigs with clear evidence of alternative code use were translated using the appropriate code tables (Supplementary Fig. 9, Supplementary Data 1, ftp://ftp.ncbi.nih.gov/pub/yutinn/crassfamily_2020/all_proteins.fa, and https://doi.org/10.5281/zenodo.4437596). Altogether, we identified 243 crAss-like phages, mostly, from the Beta, Delta, and Zeta groups, using alternative codes, with either TAG reassigned for glutamine or TGA reassigned for tryptophan (Supplementary Data 1). In most cases, stop codon reassignment was observed only in parts of the genome that encompass putative late genes. CrAss-like phages apparently can switch genetic codes at short phylogenetic

distances; for example, different branches within the Zeta group reassign either TAG or TGA or use the standard code (Supplementary Fig. 6).

One of the common mechanisms for codon reassignment employs suppressor tRNAs that have an anticodon complementary to one of the standard stop codons and are charged with an amino acid. Many genomes of crAss-like phages, especially, in the beta and zeta groups, encode multiple tRNAs including putative suppressors (Supplementary Fig. 9 and Supplementary Data 1). However, the evidence of codon reassignment is not in a perfect agreement with the presence of suppressor tRNAs (as exemplified for the zeta group in Supplementary Fig. 7), as noticed previously as well (Ivanova et al.). The absence of a detectable suppressor tRNA in phage genomes with apparent codon reassignment potentially can be explained by the use of a host-encoded suppressor or an extremely divergent phage-encoded suppressor that escapes detection with current computational methods. The converse, that is, the presence of a suppressor tRNA in the absence of apparent codon reassignment might result from incipient code change that escapes detection (see an example below).

We traced several cases of suppressor tRNA emergence. In one case, a suppressor tRNA emerges in a phage genome via a single point mutation in the anticodon of a tRNA (TTG$^{Gln}$→TTA$^{Sup}$; Supplementary Fig. 10). The block of three tRNAs (Tyr, Ley, Gln) is present in several closely related genomes from the alpha group, and one of these (OKSC01000115) carries the mutation that transforms tRNA$^{Gln}$ into a TAA stop codon suppressor. This genome as well as those of its close relatives in the alpha group show no traces of the TAA codon (or any other stop codon) reassignment. This is an apparent case of suppressor pre-emergence that would open the path for stop codon reassignment in the descendants of the mutant phage and could be a general mechanism of code switch evolution. Indeed, OFRY01000050 genome from the beta group acquired a TAG-suppressor tRNA, presumably, from a bacterial source (Supplementary Fig. 11). Unlike its closest relatives, this genome started to accumulate TAG codons in the conserved genes, such as TerL, in an apparent case of ongoing progression along the alternative coding path.

**High density of self-splicing introns and inteins in crAss-like phage genomes**. In many bacteriophages, certain genes are interrupted by Group I or Group II self-splicing introns and/or inteins[26–28]. However, some of the crAss-like phages are characterized by unprecedented density of introns and inteins. The genomes in the alpha-gamma group that have been primarily studied previously lack mobile genetic element (MGE) insertions, which enabled straightforward annotation. In contrast, phages of the Delta and, especially, Zeta groups, harbor numerous introns and inteins, which were identified by searching for the signature protein domains, primarily, endonucleases (Supplementary Fig. 12). In phages of the Delta group, group I introns are typically inserted in the core phage genes, including MCP, PolB, primase, and ATP_43b, whereas TerL typically contains an intein. The phage genomes in the Zeta group contain even more introns and inteins, up to 14 introns and 3 inteins per genome (in the OHFV01000001 cMAG). Together with the use of alternative genetic codes, the massive infestation of these genomes with introns and inteins severely complicates annotation.

An example of an intron-rich gene is the MCP gene of the Zeta group cMAG OLRF01000054 (Fig. 7a). The MCP ORF is split into five fragments across two co-directed coding frames; three of the four segments between the ORF fragments encode domains typical of Group I self-splicing introns. Within the recognizable protein-coding sequences, no standard stop codons are present, so there is apparently no alternative coding in this gene. In the

OBYQ01000140 cMAG (Delta group), TerL is encoded in three in-frame fragments that are separated by two inteins of the Hint-Hop type. Both the TerL and the intein parts of the ORF contain multiple TAG codons indicative of the use of an alternative code (Fig. 7b). Thus, in this case, intein insertion and alternative coding are combined within the same gene. PolB of the Zeta group cMAG OLWB01000021 is encoded in four fragments across three frames; two of the regions separating the coding sequences encode intron-specific domains, and both the PolB-coding and intron ORFs contain multiple TGA codons that have to be suppressed during translation (Fig. 7c).

In the Zeta group, the ORFs coding for the predicted RNAP subunits are even more drastically disrupted, with numerous stop codons and frameshifts some of which occur within highly conserved motifs, such as the DxDxD motif of RNAPB' (Fig. 4 and Supplementary Fig. 13). The predicted protein boundaries vary considerably within clades of closely related phages. It seems likely that these phages, in addition to RNA and protein splicing, and stop codon reassignment employ additional, yet unknown expression mechanisms.

## Discussion

The identification and genome analysis of the crAss-like phages described here substantially expand this group of viruses and confirm its prominence in the human gut virome, where the crAss virome accounts for about 12% of the diversity of the circular virus genomes (596 of the 4907 distinct cMAGs). However, 86.7% of the virus-specific sequence reads map to the crAss-like phage genomes, indicating that the crAss-like phage assemblage heavily dominates the human gut virome. CrAss-like phages can be readily defined as a clade and differentiated from other phages through the phylogenetic coherence of the conserved genes as well as distinct gene signatures, such as unique genes in the structural module and the giant proteins comprising the transcription apparatus. Within the CrAss-like assemblage, however, there is extensive diversity, in terms of the genome size and content as well as expression strategy. In particular, rarefaction analysis reveals an open crAss-like pangenome, so that numerous genes, expected to be involved, mostly, in the interactions between crAss-like and their Bacteroidetes hosts remain to be identified once more phage genomes are sequenced.

The originally discovered crAssphage and its close relatives[6,10,11,13] remain the most abundant viruses in the human-associated virome, but they are relatively small compared to phages in other groups, lack some characteristic genes, and therefore cannot serve as typical representatives of the crAss virome diversity. The crAss-like phages appear ripe to be formally placed into the framework of virus taxonomy. In the recently adopted multi-rank taxonomic structure[1], crAss-like phages can be expected to become an order within the existing class *Caudoviricetes* (tailed viruses of bacteria and archaea), with the Alpha-Gamma (or, possibly, Alpha and Gamma separately), Beta, Delta, Epsilon, and Zeta groups becoming the 5 (or 6) families within this order.

CrAssphage and other members of the Alpha-Gamma group described previously have comparatively simple genome organizations and, apart from the distinct transcription machinery, employ conventional strategies for genome expression. This historical accident greatly facilitated the initial genome annotation and analysis. However, this is not the case for the much broader assemblage of crAss-like phages analyzed here. In particular, many phages in the Beta, Delta, and Zeta groups employ non-standard genetic codes, and those in the Zeta group have an unprecedented high density of introns and inteins. Recoding enabled by the capture and adaptation of the suppressor tRNA can be perceived as an anti-defense strategy to impair the production of host proteins, including

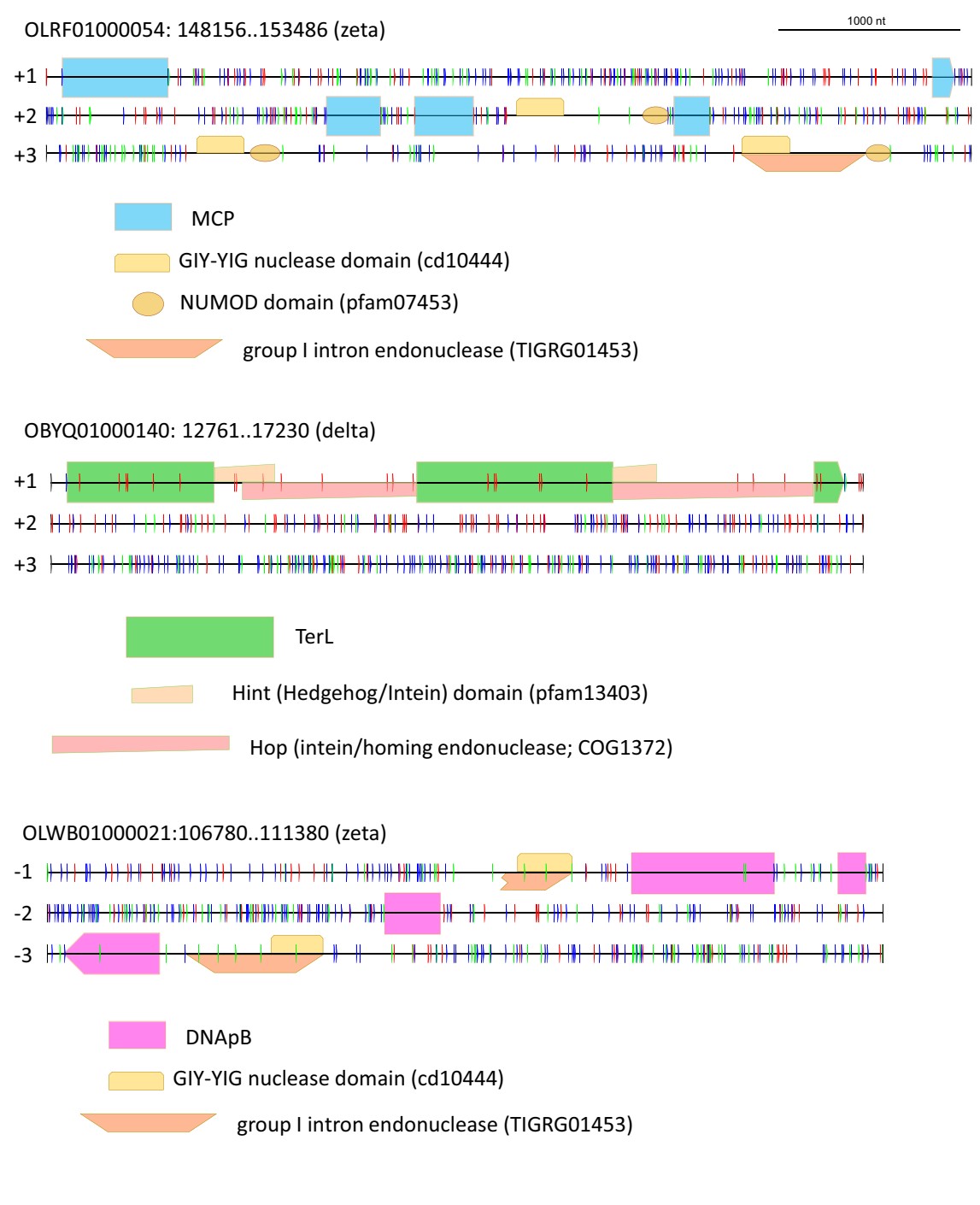

**Fig. 7 Examples of stop codon reassignments, intein, and self-splicing intron insertions in conserved genes of crAss-like phages.** Horizontal lines denote co-directed reading frames in the nucleotide sequence; short vertical bars indicate standard stop codons (red: TAG, green: TGA, blue: TAA); colored shapes indicate domains or domain fragments, mapped to the nucleotide sequence.

those involved in defense, by stop-codon readthrough, while faithfully translating the phage proteins. Whether the extreme accumulation of introns and inteins, along with potential additional, unknown forms of gene fragmentation, in the Zeta group phages is an adaptive strategy or a rampage of MGE that spun out of control due to some unknown facet of the lifestyle of these phages remains unclear. In the phylogenetic trees of conserved phage proteins, the Zeta group usually forms long branches (Fig. 1a and Supplementary Figs. 3 and 4), suggesting that evolution of the unusual features of the genome architecture of these phages is accompanied by fast

sequence evolution. Such apparent coupling between sequence and genome organization evolution rates seems to be best compatible with weak selection that makes these phages vulnerable to MGE. Clearly, however, given the high abundance of the Zeta group phages, they manage to replicate efficiently despite the genome infestation with mobile elements. Experimental investigation of these phages has the potential to reveal previously unknown mechanisms of gene expression.

How many additional crAss-like phages and, perhaps more importantly, how many distinct groups of such phages can we

expect to discover in the future? The 596 crAss-like phage genome assemblies neatly fall into five distinct groups, without many unassigned singletons, suggesting that, within the sampled metagenome diversity, the most common divisions of crAss-like phages are already known. Similarly, these 596 genomic assemblies form 221 clusters, roughly at the species level, which implies that a non-negligible fraction of crAss-like phage species has been identified. It remains to be seen how much more crAss-like phage diversity is brought about by the analysis of the vastly expanded collection of increasingly diverse metagenomes that can be anticipated to become available within the next few years.

The frequent DNAP-type switches across the entire crAss-like phage assemblage represent an enigmatic phenomenon that, to our knowledge, has not been observed previously. The pervasiveness of these switches implies a strong selective pressure that might have to do with an unknown, distinct defense mechanism characteristic of Bacteroidetes. Perhaps, such defense might involve DNAP inhibition so that DNAP-type switch becomes a means of escape. The high prevalence of in situ DNAP replacement suggests that co-regulation of genes involved in replication is important for the reproduction of crAss-like phages.

The phylogenomic study of crAss-like phages not only advances our knowledge of the human gut virome but also reveals fascinating, poorly understood aspects of phage biology. In particular, the alternative coding strategies employed by prokaryotic viruses remain to be explored in full. Along with other recent discoveries, such as the plethora of megaphages encoding enormously rich protein repertoires[24,25], these findings show that, although phages have been classic models of molecular genetics for eight decades, there is more to be learned about them than we already know.

## Methods

**Identification of cMAGs in the human gut virome.** Five thousand seven hundred and forty-two whole-community metagenome assemblies generated from human fecal samples were downloaded from the NCBI Assembly database[29] (accessed 8/2019). To limit the analyzed dataset to complete, fully assembled genomes, 95,663 "circular" contigs (50–200 bp direct overlap at contig ends) were extracted from the assemblies.

We used 421 phage-specific protein alignments, including 117 custom alignments generated previously[11] and 304 alignments from the CDD database[30], and created a set of the corresponding HMMs using hhmake[31]. Proteins in the 95,663 contigs were predicted using Prodigal[32] in the metagenomic mode and searched against the set of 421 phage-specific HMMsusing hhsearch[31], with the relaxed $e$-value cut-off <0.05. Four thousand nine hundred and seven contigs with at least one hit were selected for subsequent analysis.

**Clustering of virus genomes.** The set of 673 extended crAss-like genome assemblies identified in this work and previously[11] was searched against itself using MEGABLAST[33] with no low-complexity filtering, $e$-value threshold of $10^{-8}$, and the identity threshold for the highest scoring hit of 90%. Pairs of genomes where the coverage of the query genome by hits was at least 90% were linked, and the resulting clusters were extracted from the linkage graph. Altogether, the set formed 221 cluster; the 596 crAss-like cMAGs from the gut metagenomes and viromes belong to 169 of these clusters.

**Identification of potential hosts of the crAss-like phages.** In all, 258,077 bacterial and 4975 archaeal assemblies in the NCBI Genome database[29] were scanned for CRISPR arrays using CRISPRCasTyper[34]; this procedure yielded 3,455,966 CRISPR spacers in 236,854 arrays from 99,448 individual assemblies. A separate curated CRISPR spacer database, derived from high-quality completely sequenced genomes, containing 274,663 spacers[35], was used in parallel.

The results of the BLASTN search of 673 crAss-like genome assemblies against the spacer databases were analyzed as follows: a link between a virus assembly and a contig (genome partition) containing a CRISPR array was established when BLASTN yielded one or more hits with at least 90% of the spacer positions matching the corresponding phage sequence or two or more hits with at least 80% identity. Then the source organism of the contig with the highest scoring hit was selected as the potential host, with the preference given to spacers derived from completely sequenced genomes.

The taxonomic assignment of non-Bacteroidetes potential host contigs was verified by running a translating search against the protein database derived from

completely sequenced genomes; the lowest-level taxa, accounting for at least 75% of all hits were considered a "safe" taxonomic affiliation (e.g., JUJT01000004.1, deposited in GenBank as *Pectobacterium brasiliense*, produced best hits into complete genomes of Enterobacteriales, confirming this order as the most probable taxonomic assignment for this contig).

**Estimation of phage abundances in the human gut virome.** To estimate the abundances of the candidate phages in metagenomic samples, a set of 500 human gut metagenomic samples was randomly selected from the SRA database. The amplicon-based and RNA-seq metagenomes were removed, resulting in 425 samples (see Supplementary Material). Adapters and technical sequences were removed by Trimmomatic v0.39 using all supplied adapter sequences[36]. To estimate the abundances, the reads from each sample were aligned against a database that consisted of all complete bacterial, archaeal, and viral genomes in RefSeq, with the addition of the crAss-like sequences identified in the present work. The sequences from phages previously included in the clade "unclassified Podoviridae" (which includes the crAss-like virus clade) were removed to avoid duplications of the reference sequences that could skew the abundance estimates.

Kraken2 was used to align reads from each sample against the constructed database[37]. Reads that aligned to multiple database entries were assigned to their last common ancestor in the taxonomy tree. Reads assigned to internal taxonomical nodes were reassigned by Braken, using the number of reads from all samples with unique mappings to the database[38].

**Gene calling with alternative genetic codes and tRNA scan.** A modified version of the Prodigal program[32], which allows assigning amino acids to codons that serve as stop codons in the standard bacterial genetic code (-g 11), was used to translate the genes with the suspected stop codon reassignment. All contigs were searched for the presence of tRNAs using tRNA-scan-SE (v. 2.0)[39] employing a bacterial model of tRNAs (-B) and bitscore cut-off of 35.

**Protein sequence analysis.** A combination of PSI-BLAST[40] searches using the CDD database[30] profiles and custom profiles as queries and HHPRED[31] searches was used for domain identification and functional annotation of proteins. Protein annotations of 110 representative genomes are available at ftp://ftp.ncbi.nih.gov/pub/yutinn/crassfamily_2020/prot2profile.txt and at https://doi.org/10.5281/zenodo.4437596.

Multiple sequence alignments of protein sequences were constructed using MUSCLE[41]. Phylogenetic reconstruction was performed using the IQ-TREE program, with the evolutionary models selected by IQ-TREE[42].

Detection of cMAG homologs was performing by running BLASTP search of predicted proteins against the subset of proteins in the NCBI NR database with taxonomic label "Viruses", $e$-value threshold of 0.0001, and no low complexity filtering.

Sequences from representative genomes not annotated with any profiles were clustered using MMSEQ2[43] with the similarity threshold of 0.4.

**Rarefaction analysis of the crAss-like pangenome.** Rarefaction analysis of the crAss-like phage pangenome was performed by constructing 1001 random-ordered lists of the 110 representative genomes and counting the total number of distinct (differently annotated) genes in the first $k$ genomes in each list. Profile classes were used as gene annotations when available, and cluster IDs were used otherwise; unclustered genes were counted as singletons.

**Reporting summary.** Further information on research design is available in the Nature Research Reporting Summary linked to this article.

## Data availability

The data were collected from public databases: https://www.ncbi.nlm.nih.gov/assembly (5742 whole-community metagenome assemblies generated from human fecal samples; see Supplementary Data 2 for the list) CDD, https://www.ncbi.nlm.nih.gov/cdd (CD, PFAM, and GOG profiles); NCBI Genome database, https://www.ncbi.nlm.nih.gov/genome (bacterial and archaeal genomes as of July 2020); and GenBank, https://www.ncbi.nlm.nih.gov/Genbank/ (whole GenBank database was searched); SRA, https://trace.ncbi.nlm.nih.gov/Traces/sra/ (a random sample of sets, see Supplementary Data 2 for the list). All data generated in the course of this work is available at ftp://ftp.ncbi.nlm.nih.gov/pub/yutinn/crassfamily_2020/ and at https://doi.org/10.5281/zenodo.4437596 without restrictions. Source data are provided with this paper.

## Code availability

All custom software used in this work is available from: ftp://ftp.ncbi.nlm.nih.gov/pub/yutinn/crassfamily_2020/ and at https://doi.org/10.5281/zenodo.4437596.

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

## Acknowledgements

The authors thank Ayal Gussow for help with the prediction of Acrs, Kira Makarova for help with sequence analysis, and Koonin group members for useful discussions. N.Y., S.B., S.A.S., Y.I.W., and E.V.K. are supported by the Intramural Research Program of the National Institutes of Health of the USA (National Library of Medicine); P.A.P. is supported by the NSF/MCB-BSF grant 1715911.

## Author contributions

E.V.K and P.A.P. initiated the project; M.R. and D.A. collected the data and performed the initial data analysis; N.Y., S.B., S.A.S., Y.I.W., and I.T. performed the detailed data analysis; N.Y. and E.V.K. wrote the manuscript that was read, edited, and approved by all authors.

## Funding

## Competing interests

The authors declare no conflict of interest.
