## [Peer Review File · Nature Communications]

REVIEWER COMMENTS

Reviewer #1 (Remarks to the Author):

In the last years, great attention has been devoted to the study of CrAssPhage. In this paper, the authors expanded the current knowledge of the viruses belonging to the crAssphage group, showing some interesting features of their genomes. In addition, two new groups of crAss-like phages were identified, shading further light on the evolutionary history of this group. The paper contains a remarkable evidence of the possibility that alternative genetic codes can emerge even at short phylogenetic distances, suggesting that alternative translation tables should be taken into account in these kinds of studies. The authors also provide insights in the genome architecture of crAssphage-like viruses, pointing out some interesting differences between groups. Overall, these crAssphage-like genomes are proposed to become a new order divided into 5 or 6 subgroups. While the authors provide some degree of evidence for this, wider analyses are needed to confirm that crAssphages can be considered a separate order from Caudovirales.

The analysis is limited to fully assembled circular genomes. However, one cannot exclude the presence of well reconstructed genomes that, in some circumstances, can lack the circularized part (in this way, contigs of good quality that correspond to nearly complete genomes are excluded). Nevertheless, the authors reconstructed nearly 600 genomes of crAssphage-like viruses.

Finally, the authors assessed the abundance of the retrieved crAssphage genomes in gut metagenomes. The paper is mostly centred on the genomic characterization of such novel genomes, so the part on the prevalence and abundance of crAssphage is not as comprehensive as it could be. While there is the potential to expand this aspect (see below), it is not probably key given the focus on the genetic/genomic aspects that are indeed novel and interesting.

Overall, the paper is well written and represents a relevant contribution to the knowledge of crAssphages, even though some points should be better clarified.

Main points/suggestions

1. The genomes analyzed come from different assemblies, in which a plethora of different techniques were used. Could this heterogeneity constitute a problem and lead to artifacts? Also, could the different experimental techniques influence the retrieval of circular genomes?

2. The authors state that the evidence of codon reassignment is not in perfect agreement with the presence of suppressor tRNAs and then suggest the possible existence of alternative meanings for genetic code fine-tuning (page 11, lines 9-11). However they do not propose any alternative mechanism for this. A further speculation on this point would be appreciated.

3. CRISPR arrays have indeed proven to be useful to reconstruct the host-association of phages. However their presence doesn't guarantee that the phage infects that particular host (e.g., old relationships between the phage and the host or an acquisition of the spacer through HGT or recombination may lead to false positives, while the absence of the spacer does not ensure negative associations). Are there any other tools or approaches for host prediction that can be employed here (in conjunction with the CRISPR)? This would help to ensure that almost all of the newly retrieved crAssphages are associated with Bacteroides.

4. The authors report a "substantial within-group diversity" in the retrieved crAssphage genomes of the Epsilon and Zeta group (page 6). However, there is no quantitative statement for this. It would be interesting to compare the nucleotide diversity within and between groups (e.g. ANI). This would allow to better explore if there is a continuum of genetic diversity among the proposed groups. This would surely be an important confirmation of the relatedness of the retrieved genomes with available crAssphage sequences.

5. At page 6, the authors say that all the crAss-like genomes share 8 genes in the structural block. However, phylogenetic analysis is shown for the large terminase subunit (TerL), DNA primase and DNA polymerase only. A nice complementary confirmation to the division of crAssphage-like genomes into the 5-6 proposed groups could come from comparing phylogenies of conserved proteins (e.g. portal, MCP, gene75, gene 74...). This has been done, for example, in Guerin et al. (Cell Host & Microbe, 2018). If the phylogenetic signal confirms the group-assignments on multiple (shared) genes, this would improve the analysis.

6. Do the proteins that cannot be annotated (i.e. the white proteins in figure 2) share any similarity with each other? Is there any "non annotated" conserved protein between those crAssphages? Should this be the case, it would also help to confirm that genomes in the Epsilon- and Zeta- are related with the original crAssphages in the Alpha- and Delta- groups.

7. The evaluation of different crAssphage-like genomes abundance in the human gut is an essential part of the work, since it is also reprised by the title. However, the authors only checked 500 metagenomes (which then become 425 after removing 16S rRNA gene amplicon sequencing and RNA-Seq samples). This number can be increased to further ensure that crAss-like phages are "the

dominant component of the human gut virome". Given the focus of the paper, this point should be considered as an optional suggestion. On the other hand, it is very important that the human gut metagenomes employed should be available as SRA accession numbers.

8. Also regarding crAssphage-like genomes abundance, the authors say in the methods that they used a custom Kraken2 database with the sequences of all the crAss-phage like genomes. The authors say that: "Among the sequence read mapped to putative virus genomes (see Methods for details), crAss-like phages accounted for 86.7%". Is this based on the reads mapping against viruses only? In other words, are the authors re-normalizing the percentages of Figure 4 to the fraction of reads that match to a virus? This should be better clarified (maybe in the figure caption). For completeness, real abundances should also be given (i.e. the real percentage of reads aligning to each of the 5-6 crAssphage subgroups). The authors should also report the prevalence of each crAssphage-group in the explored cohort.

Minor points and typos

- Page 3, line 5: the authors refer to the human virome as the collection of DNA sequences from virus-like particles. This should be rephrased to encompass the whole complexity of the human virome, so to include also RNA viruses.

- Page 3: could the authors explain the role of RNAP in crAssphage transcription in better detail? I think this would be of help to the reader.

- Page 3, lines 28-29: add a comma after "crAss-like phages belong to the bacterial phylum Bacteroidetes"

- Page 4: "This set of cMAGs presumably should consist of plasmid and virus genomes.". This sentence needs clarification. How can the authors be sure of this? If this is due to the later HMM-based filtering of the cMAGs to identify viral sequences, this sentence should be moved below or rephrased, as not all cMAGs are sure to be viral (or plasmids).

- Page 4: "For the detected TerL homologs, phylogenetic analysis revealed a major, strongly supported clade that included 596 crAss-like phage cMAGs which formed 169 clusters of distinct genomes sharing less than 90% of similarity at the DNA level which roughly corresponds to the level of virus species". The authors should point the reader to the methods, as the clustering is explained

there. Also, the concept of different viral species diverging for more than 90% is not universal; the authors should add something like “we intend as different viral species...” or similar.

- Page 6, line 3: “... subsequently substantially ...”; reword this phrase.

- Page 6, line 5: “... Epsioln -> Epsilon ...”

- Page 7, line 14: Remove the comma after “As this analysis shows, some of the crAss-like phages, particularly”

- Page 7, line 24: remove “of” after “The recent structural and functional study of the RNAP of Cellulophaga baltica phage phi 14:2”

- Page 8, line 7: “... anti-CRSPR -> anti-CRISPR”

- Page 11, line 9: “... prefect -> perfect”

- Page 14, line 24: remove the “.” after “custom alignments generated previously”

- Page 16, line 7: “... the -> that”

- In the figure legends of Figure 1, S2 and S3 should be clarified what the leaves color mean in the phylogenetic trees.

- The authors should specify where the 5742 assemblies come from (e.g., by providing the NCBI research query or a complete list of accessions).

Reviewer #2 (Remarks to the Author):

Yutin et al. add to the growing list of papers describing the ubiquitous crAss-like phages. They describe 600 additional crAss-like phage genomes and reveal several genomic features and diversity that were not previously appreciated. The main conclusions of the paper seem to rest on the idea that with more sampling comes greater diversity and suggests that we are nowhere near a complete understanding of the crAss-like phage group. The authors highlight some potentially interesting features for study in future work. However, I struggled to understand what the key takeaway of the work was beyond that “there are many assembled crAss-like phages and they are diverse.” Perhaps the authors can revise the paper to provide deeper insights into the following:

- 1) Identify crAss-like phage genome features associated with host disease status. For example, previous work showed that IAS virus is enriched in individuals with late-stage AIDS (Oude Munnink et al 2014 BMC Infectious Disease)

- 2) Conduct rarefaction analysis (e.g. Pope et al 2015 Elife) on all described crAss-like phage genomes to understand gene flux through the family and how close we are to a saturated sampling of crAss-like phages.

- 3) Determine whether the newly identified crAss-like phages share high degrees of sequence identities with other phages (e.g. are there modules of genes that are similar to those from other previously characterized Bacteroides phages such as Hankeyphage, B40-8, etc.)

Reviewer #3 (Remarks to the Author):

Yutin and colleagues present here an in-depth genomic characterization of newly identified crAss-like phages. They propose the creation of two new families and highlight unusual genomic features that contribute to the challenge of annotating this group of phages. The genomic characterization is very exhaustive, and the findings are well explained. The proposed use of alternative genetic codes for some of these phages is quite interesting. Although, it is still unclear what are the biological roles of these new genomic features.

1. L29-30: These anti-defense strategies are still highly speculative. Should not be in the abstract but in the discussion.
2. L46: While I get the point from the bioinformatic side, this genome is not circular in the phage capsid as it would not be ejected from the capsid. Therefore, from a phage biology perspective, it seems to bring unnecessary confusion as written.
3. L85: Why do cMAGs presumably consist only of plasmids and virus genomes?
4. L90: 4,907 out of 95,663 contigs (5.1%)?
5. L91-92: Ref? Table? Data not shown?
6. L93/L111: 596 or 595 (Fig 1B)?
7. L95: Provide the exact number of previously identified crAss-like phages used in your study.
8. L94-95: Do you have a reference for the 90% similarity? This may higher than 90%? Besides, should it be identity or similarity?
9. L102: You could consider exploring other host prediction methods for the 207 phages that lack a predicted host using CRISPR spacers. Also, did you find a relationship between the phage family and the bacterial host?
10. L102: Not sure where these “673 phages” are coming from?
11. L105: What is a reliable spacer match?
12. L106: Replace “the” with “than”.
13. L107: I did not have access to Supplementary Tables S1 and S2, so I was unable to give comments. It would be of interest to the readers to write some of these bacterial phyla in the text and perhaps comments on the meaning of these potential additional hosts.
14. L111: 596-strong? Meaning?
15. L120: Reference for this phage? See *Journal of Aquatic Animal Health* 31:225–238, 2019. *Flavobacterium*-infecting phage Fpv3 seems to belong to the (current) Podoviridae family. crAss phages are apparently also podophages. As far as I know *Flavobacterium* do belong to the Bacteroidetes phylum. Are *Flavobacterium* found in the human gut? If yes... Still, I see the point of excluding this one (one out of 596). Would this suggest a food origin?
16. L120: Does *Flavobacteroides* exist?
17. L129: Replace “Epsioln” with “Epsilon”.
18. L129-132: Considering that Cellulophaga and Methylophaga hosts are mostly found in marine/sea environments, does this suggest that these Epsilon crAss phages may not be native human gut phages?

19. L132-138: Obviously, such larger genome would have to fit in the phage capsid. Any significant difference in the morphogenesis capsid/head genes/proteins? With such larger genomes, one may argue/wonder if Epsilon and Zeta should still be considered crAss-like phages?
20. L139-141: Would this be expected considering that most phages have similar discernible block of genes and morphogenesis genes are often conserved in related phages.
20. L151: Replace "all crAss-like genes" with "all crAss-like phages".
21. L170: seems to possess?
22. L173: Epsilon.
23. L178: Replace "phi 14:2 of confirms" with "phi 14:2 confirms".
24. L183: "many phages". Do these include podophages?
25. L185: Indeed, the presence of such nuclease in the structural block seems odd. Is this common in bacterial viruses or is limited to a group of phages?
26. L190: One would assume that the targeting spacers were in arrays associated to specific types/subtypes of CRISPR-Cas systems?
27. L193-195: Are there other phages for which unknown orfs were proven to target host defense systems?
28. L208-239: While the case for two DNAPs has been elegantly demonstrated, it is still unclear why such switching/displacement occurred in the first place and its role in the biology of crAss-like phages?
29. L229-230: Replace "appears to have involved" with "appears to have been involved".
30. L241: This is a really interesting section. Wish it could have been confirmed by some proteomic work. Obviously can't for now.
31. L250/L254: Some? How many? Are the numbers in Fig. 1B? Seems really unique to Delta and Zeta groups.
32. L263-265: Any example with characterized phages (instead of only metagenomic analyses)?
33. L270: Beta? They don't seem to have supp tRNA (Fig. 1B).
34. L273: late genes as late-expressed genes? I would assume these are putative late-expressed genes? Any reason why this would be only for structural proteins? Presumably because it needs time to produce those suppressor tRNAs and translate a significant amount of structural proteins?
35. L280: prefect? perfect?
36. L284: Replace "emerges in one a phage genome" with "emerges in a phage genome".
37. L295: Any hypothesis on why such alternative coding path?

38. L297/309: Infestation? Consider revising.
39. L299: Group II introns have been found in phages? Reference would be needed for this.
40. L307: Replace “Zzeta” with “Zeta”.
41. L310: Not only does it severely complicate annotation but I also wonder how it would impact their fitness? Any other characterized phages with such relatively high number of introns, either in the mcp or terL or rnap?
42. L311: Define MCP and it should be mcp in italic.
43. L333: I did not find any elements in the Results section regarding the “12% of the diversity” mentioned in the Discussion.
44. L355-358: Are there proofs of such anti-defense strategy in the literature?
45. L384: Remove “.” after ref 11, replace “workand” with “work and”
46. L390: Remove “.” after ref 11.
47. L425: Replace “database the consisted” with “database that consisted”. Also in your Results sections, there is no mention of bacterial and archaeal abundance, however your databased included bacterial and archaeal genomes.
48. L346: Caudovirecetes? Reference? Google and PubMed searches revealed no hit (except your bioRxiv manuscript)?
49. L345-348: If they all become families, one makes you wonder if they will still be called like crAss-like phages?
50. L351-352. As written, this seems to reduce the impact of the previous work? Consider revising or remove.
51. L356-358: Not very convincing. Defense strategies would already be in place when phage infects the cell. However, it could be used to redirect the host machinery to produce phage proteins.
52. L359: Still not sure how such a massively intron/intein-containing phage would be functional or efficiently replicate (fitness costs, etc).
53. L365: Could this be due to the host they are infecting and not related to a defense mechanism?
54. References: Are bioRxiv papers accepted (see ref 29)?
55. Figure 1: Replace “shaded” with “highlighted”.
56. Figures 1 and 2 are well-designed and easy to understand.
57. Figure 3: Why did you decide to combine results for the transcription gene block and for the stop codons in the same figure? I don’t get the main message of this Figure at the first glance. I think it would be best to remove the stop codon locations from this figure, since it is represented in Figure 6.

58. The authors should consider adding Supp Fig. 7 to the manuscript.

Yutin et al

Response to the reviewers

REVIEWER COMMENTS

Reviewer #1 (Remarks to the Author):

In the last years, great attention has been devoted to the study of CrAssPhage. In this paper, the authors expanded the current knowledge of the viruses belonging to the crAssphage group, showing some interesting features of their genomes. In addition, two new groups of crAss-like phages were identified, shading further light on the evolutionary history of this group. The paper contains a remarkable evidence of the possibility that alternative genetic codes can emerge even at short phylogenetic distances, suggesting that alternative translation tables should be taken into account in these kinds of studies. The authors also provide insights in the genome architecture of crAssphage-like viruses, pointing out some interesting differences between groups. Overall, these crAssphage-like genomes are proposed to become a new order divided into 5 or 6 subgroups. While the authors provide some degree of evidence for this, wider analyses are needed to confirm that crAssphages can be considered a separate order from Caudovirales.

The analysis is limited to fully assembled circular genomes. However, one cannot exclude the presence of well reconstructed genomes that, in some circumstances, can lack the circularized part (in this way, contigs of good quality that correspond to nearly complete genomes are excluded). Nevertheless, the authors reconstructed nearly 600 genomes of crAssphage-like viruses.

Yes, given the abundance of complete, circularizable genomes, contigs that lack direct terminal repeats fell out of the scope of the current study. We have not reconstructed these genomes but rather downloaded them from GenBank where they were already present in the complete form.

Finally, the authors assessed the abundance of the retrieved crAssphage genomes in gut metagenomes. The paper is mostly centred on the genomic characterization of such novel genomes, so the part on the prevalence and abundance of crAssphage is not as comprehensive as it could be. While there is the potential to expand this aspect (see below), it is not probably key given the focus on the genetic/genomic aspects that are indeed novel and interesting.

Overall, the paper is well written and represents a relevant contribution to the knowledge of crAssphages, even though some points should be better clarified.

Main points/suggestions

1. The genomes analyzed come from different assemblies, in which a plethora of different techniques were used. Could this heterogeneity constitute a problem and lead to artifacts? Also, could the different experimental techniques influence the retrieval of circular genomes?

Potential assembly errors, certainly, represent a concern for all follow-up studies of viral genomes. Specifically, chimeric reads (that typically account for up to 1% of all reads in Illumina datasets) and errors in repeat resolution can result in chimeric genomes formed by fusions of segments from different viruses. However, the vast majority of datasets in our study were assembled using the SPAdes/metaSPAdes assembler that was originally designed for assembling single-cell datasets with a high level of chimerism (Bankevich et al., JCB 2012). The analysis of chimeric reads was further improved in the follow-up work (Nurk et al., JCB 2013), and extensive benchmarking has shown that SPAdes rarely generates chimeric contigs. Similarly, SPAdes relies on an accurate and extensively benchmarked repeat resolution module that is unlikely to generate chimeric contigs (Prjibelsky et al. Bioinformatics 2014). Given these safeguards incorporated in SPAdes, assembly artifacts in circular contigs appear to be highly unlikely.

2. The authors state that the evidence of codon reassignment is not in perfect agreement with the presence of suppressor tRNAs and then suggest the possible existence of alternative meanings for genetic code fine-tuning (page 11, lines 9-11). However they do not propose any alternative mechanism for this. A further speculation on this point would be appreciated.

We agree that such discussion is useful and included a brief speculation along these lines (ln 320-325)

3. CRISPR arrays have indeed proven to be useful to reconstruct the host-association of phages. However their presence doesn't guarantee that the phage infects that particular host (e.g., old relationships between the phage and the host or an acquisition of the spacer through HGT or recombination may lead to false positives, while the absence of the spacer does not ensure negative associations). Are there any other tools or approaches for host prediction that can be employed here (in conjunction with the CRISPR)? This would help to ensure that almost all of the newly retrieved crAssphages are associated with Bacteroides.

To our knowledge, all other approaches to phage-host pairing are less reliable than CRISPR analysis. Nevertheless, we compared all phage-encoded proteins from our dataset to the database of proteins from completely sequenced prokaryotic genomes (to make sure that the genome attribution of proteins is reliable). These searches yielded strong matches for 516 phage proteins (coverage by length >2/3; identity >50%) in prokaryotic genomes, suggestive of recent gene exchange. Thus, this gene set should be expected to be enriched with genes acquired from the hosts. We found that 325 of these proteins (63%) match Bacteroidetes sequences, a fraction that is 11 times greater than expected by chance (Bacteroidetes proteins comprise 6% of the database). We added this information to the "prok matches" sheet of Table S1 and described it in the revised text (ln 132 -139).

4. The authors report a “substantial within-group diversity” in the retrieved crAssphage genomes of the Epsilon and Zeta group (page 6). However, there is no quantitative statement for this. It would be interesting to compare the nucleotide diversity within and between groups (e.g. ANI). This would allow to better explore if there is a continuum of genetic diversity among the proposed groups. This would surely be an important confirmation of the relatedness of the retrieved genomes with available crAssphage sequences.

The extent of diversity among these phages makes ANI comparisons uninformative and effectively irrelevant because the similarity at the nucleotide level barely exceeded the random expectation. We performed the relative tree depth analysis using the four conserved genes (TerL, MCP, portal and gene75) to compare the intra- and inter-group phylogenetic distances and incorporated the results in the text (ln 157-177) and the Supplementary data (sheet “depth” in Supplementary Table S1).

5. At page 6, the authors say that all the crAss-like genomes share 8 genes in the structural block. However, phylogenetic analysis is shown for the large terminase subunit (TerL), DNA primase and DNA polymerase only. A nice complementary confirmation to the division of crAssphage-like genomes into the 5-6 proposed groups could come from comparing phylogenies of conserved proteins (e.g. portal, MCP, gene75, gene 74...). This has been done, for example, in Guerin et al. (Cell Host & Microbe, 2018). If the phylogenetic signal confirms the group-assignments on multiple (shared) genes, this would improve the analysis.

Concatenating alignments imposes a strong (even if often left implicit) assumption that the histories of all genes are congruent. We believe that a curated alignment of a single “well-behaved” gene (TerL in our case) is sufficiently informative to represent the overall structure of the group. This said, we constructed three other trees (MCP, portal and gene75) for comparison (ftp://ftp.ncbi.nih.gov/pub/yutinn/crassfamily_2020/trees); they all support the monophyly of each of the 5 major groups of crAss-like phages even if the details of the phylogenies vary. This is mentioned in the revised manuscript (ln 154-156).

6. Do the proteins that cannot be annotated (i.e. the white proteins in figure 2) share any similarity with each other? Is there any “non annotated” conserved protein between those crAssphages? Should this be the case, it would also help to confirm that genomes in the Epsilon- and Zeta- are related with the original crAssphages in the Alpha- and Delta- groups.

Yes, there are annotated proteins that are group-specific or even shared among two or more groups but not universally conserved that were left white to avoid cluttering the figure with multiple colors. The list of 6,440 protein-to-profile associations (de facto grouping proteins into families) is available at the FTP site ([prot2profile.txt at ftp://ftp.ncbi.nih.gov/pub/yutinn/crassfamily_2020/](ftp://ftp.ncbi.nih.gov/pub/yutinn/crassfamily_2020/)).

7. The evaluation of different crAssphage-like genomes abundance in the human gut is an essential part of the work, since it is also reprised by the title. However, the authors only checked 500 metagenomes (which then become 425 after removing 16S rRNA gene amplicon sequencing and RNA-Seq samples). This number can be increased to further ensure that crAss-like phages are “the dominant component of the human gut virome”. Given the focus of the paper, this point should be considered as an optional suggestion. On the other hand, it is very important that the human gut metagenomes employed should be available as SRA accession numbers.

Analysis of SRA data is computationally expensive; substantially expanding it is beyond our capability within a reasonable time frame. The 425 SRA accession numbers that were used in the analysis are listed in the sheet “run accessions” of the Supplementary Table S2.

8. Also regarding crAssphage-like genomes abundance, the authors say in the methods that they used a custom Kraken2 database with the sequences of all the crAss-phage like genomes. The authors say that: “Among the sequence read mapped to putative virus genomes (see Methods for details), crAss-like phages accounted for 86.7%”. Is this based on the reads mapping against viruses only? In other words, are the authors re-normalizing the percentages of Figure 4 to the fraction of reads that match to a virus? This should be better clarified (maybe in the figure caption). For completeness, real abundances should also be given (i.e. the real percentage of reads aligning to each of the 5-6 crAssphage subgroups). The authors should also report the prevalence of each crAssphage-group in the explored cohort.

The full data on the abundance analysis is provided in the Supplementary Table S2, sheet “abundances”.

Minor points and typos

- Page 3, line 5: the authors refer to the human virome as the collection of DNA sequences from virus-like particles. This should be rephrased to encompass the whole complexity of the human virome, so to include also RNA viruses.

This line was corrected to “the nucleotide sequences from the virus-like particle fraction”, thus covering the RNA virome.

- Page 3: could the authors explain the role of RNAP in crAssphage transcription in better detail? I think this would be of help to the reader.

In the revised manuscript (ln 64-66), we expanded on the functional characterization of the crAss-like phage RNAP reported in Ref. 12: <https://www.biorxiv.org/content/10.1101/2020.03.07.982082v1>

- Page 3, lines 28-29: add a comma after “crAss-like phages belong to the bacterial phylum Bacteroidetes”

added

- Page 4: “This set of cMAGs presumably should consist of plasmid and virus genomes.”. This sentence needs clarification. How can the authors be sure of this? If this is due to the later HMM-based filtering of the cMAGs to identify viral sequences, this sentence should be moved below or rephrased, as not all cMAGs are sure to be viral (or plasmids).

Generally, cMAGs are expected to be viruses or plasmids, but artifacts are possible, obviously. The sentence has been modified to say “presumably contain” instead of “should consist of”.

- Page 4: “For the detected TerL homologs, phylogenetic analysis revealed a major, strongly supported clade that included 596 crAss-like phage cMAGs which formed 169 clusters of distinct genomes sharing less than 90% of similarity at the DNA level which roughly corresponds to the level of virus species”. The authors should point the reader to the methods, as the clustering is explained there. Also, the concept

of different viral species diverging for more than 90% is not universal; the authors should add something like “we intend as different viral species...” or similar.

Corrected as suggested.

- Page 6, line 3: “... subsequently substantially ...”; reword this phrase.

Corrected (‘substantially’ has been removed)

- Page 6, line 5: “... Epsioln -> Epsilon ...”

Corrected

- Page 7, line 14: Remove the comma after “As this analysis shows, some of the crAss-like phages, particularly”

Corrected

- Page 7, line 24: remove “of” after “The recent structural and functional study of the RNAP of Cellulophaga baltica phage phi 14:2”

Corrected

- Page 8, line 7: “... anti-CRSPR -> anti-CRISPR”

Corrected

- Page 11, line 9: “... prefect -> perfect”

Corrected

- Page 14, line 24: remove the “.” after “custom alignments generated previously”

Corrected

- Page 16, line 7: “... the -> that”

Corrected

- In the figure legends of Figure 1, S2 and S3 should be clarified what the leaves color mean in the phylogenetic trees.

Corrected

- The authors should specify where the 5742 assemblies come from (e.g., by providing the NCBI research query or a complete list of accessions).

As suggested, the list was added to Supplementary Table S2, sheet “assemblies accessions”. The NCBI search query was: [https://www.ncbi.nlm.nih.gov/assembly/?term=txid408170\[Organism:noexp\]](https://www.ncbi.nlm.nih.gov/assembly/?term=txid408170[Organism:noexp])

At the time of our study, there were 5742 assemblies matching this query.

Reviewer #2 (Remarks to the Author):

Yutin et al. add to the growing list of papers describing the ubiquitous crAss-like phages. They describe 600 additional crAss-like phage genomes and reveal several genomic features and diversity that were not previously appreciated. The main conclusions of the paper seem to rest on the idea that with more sampling comes greater diversity and suggests that we are nowhere near a complete understanding of the crAss-like phage group. The authors highlight some potentially interesting features for study in future work. However, I struggled to understand what the key takeaway of the work was beyond that “there are many assembled crAss-like phages and they are diverse.” Perhaps the authors can revise the paper to provide deeper insights into the following:

1) Identify crAss-like phage genome features associated with host disease status. For example, previous work showed that IAS virus is enriched in individuals with late-stage AIDS (Oude Munnink et al 2014 BMC Infectious Disease)

Unfortunately, the metadata tags in database entries do not allow for a straightforward meaningful analysis (only a few are clearly associated with the health of the sampled individuals). In addition, recent research (<https://doi.org/10.1038/s41564-019-0494-6>) suggests that the apparent associations could be confounded. Our understanding is that this issue needs to be studied as a separate project, rather than in passing, which is the best treatment we can provide within the scope of this work.

2) Conduct rarefaction analysis (e.g. Pope et al 2015 Elife) on all described crAss-like phage genomes to understand gene flux through the family and how close we are to a saturated sampling of crAss-like phages.

Rarefaction analysis (random independent subsampling of genomes) works only under the assumption that the tree of the analyzed genomes is a star. This is not true of the Pope et al. work (see their Figure 3), and is absolutely not the case with crAss-like phages (see any tree in the present work). When genomes are related to each other through a non-trivial tree, they are not independent and the sampling density does not directly reflect the information content of the sample (the more clustered the genomes, the fewer of them is needed to recover most of the pangenome). With a dataset like crAss-like phages, rarefaction analysis is bound to produce highly misleading results. The proper way to study gene flux would involve constructing gene exchange models (see e.g. <https://doi.org/10.1093/gbe/evs016> or <https://doi.org/10.1073/pnas.1614083113>), but this, again, would be well beyond the scope of this work.

3) Determine whether the newly identified crAss-like phages share high degrees of sequence identities with other phages (e.g. are there modules of genes that are similar to those from other previously characterized Bacteroides phages such as Hankeyphage, B40-8, etc.)

We performed an all-vs-all comparison of the predicted ORFs encoded by crAss-like phages with all ORFs encoded by other phages in GenBank (including the 27 Bacteroides phage genomes recently released). This analysis confirmed the findings of Hryckowian and Merrill et al 2020 that phages DAC15 and DAC17

are crAss-like phages, and we updated our results section with this information in line 150. Further, we could identify a syntenic block of 5 genes encoded by a Zeta family crAssphage that is present in two blocks in DAC16, another Bacteroides-infecting phage that is not part of the extended crAss-like assemblage. This finding is mentioned in lines 162-165 and is illustrated by the new Supplementary Figure XX. Overall, 3001 non-crAss-like phages possess at least one ORF with significant sequence similarity (>27% amino acid identity over >50% length) to at least one crAss-like phage. However, no large syntenic blocks of highly similar genes were detected between crAss-like phages and any phages outside this assemblage.

Reviewer #3 (Remarks to the Author):

Yutin and colleagues present here an in-depth genomic characterization of newly identified crAss-like phages. They propose the creation of two new families and highlight unusual genomic features that contribute to the challenge of annotating this group of phages. The genomic characterization is very exhaustive, and the findings are well explained. The proposed use of alternative genetic codes for some of these phages is quite interesting. Although, it is still unclear what are the biological roles of these new genomic features.

1. L29-30: These anti-defense strategies are still highly speculative. Should not be in the abstract but in the discussion.

We followed the reviewer's suggestion and dropped the speculation from the Abstract.

2. L46: While I get the point from the bioinformatic side, this genome is not circular in the phage capsid as it would not be ejected from the capsid. Therefore, from a phage biology perspective, it seems to bring unnecessary confusion as written.

Adjusted as suggested: "appears to be circular in sequence analysis (by analogy with other phages with pseudo-circular genomes, this is, probably, a terminally redundant linear genome)".

3. L85: Why do cMAGs presumably consist only of plasmids and virus genomes?

Presumably, cMAGs represent circular (plasmids) or pseudo-circular (viruses) genomes. Obviously, artifacts are possible. We corrected the sentence in question to: "This set of cMAGs presumably contains a high fraction of plasmid and virus genomes".

4. L90: 4,907 out of 95,663 contigs (5.1%)?

The rest is composed of plasmids, unidentified viruses, and possible assembly artifacts. The criteria are clear as stated in the manuscript.

5. L91-92: Ref? Table? Data not shown?

A reference to Supplementary Table S1 added

6. L93/L111: 596 or 595 (Fig 1B)?

596 crAss-like cMAGs were detected. As explained in the Results section, “One cMAG did not belong to any of these 5 clades but rather grouped with the Flavobacterium phage Fpv3 (a phage infecting a fish pathogenic bacterium); this might not be a native human gut phage”. Thus, the 5 groups account for 595 of the 596 cMAGs.

7. L95: Provide the exact number of previously identified crAss-like phages used in your study.

70 previously identified phage genomes were included in the phylogenetic reconstructions in this work (Table S1). However, 23 of these (from Ivanova et al 2014) were not identified as crAss-like at the time.

8. L94-95: Do you have a reference for the 90% similarity? This may higher than 90%? Besides, should it be identity or similarity?

As described in Methods, we define 90% genome similarity if/when the two genomes are at least 90% identical on at least 90% of genome length. Given this criterion, the term ‘similarity’ appears more appropriate.

9. L102: You could consider exploring other host prediction methods for the 207 phages that lack a predicted host using CRISPR spacers. Also, did you find a relationship between the phage family and the bacterial host?

Other host prediction methods are not as reliable as CRISPR analysis. Nevertheless, as pointed out in the response to reviewer 1, in the revised manuscript, we additionally included the search for strong similarity between phage and bacterial proteins that is likely indicative of gene exchange, and hence, of the host identity.

10. L102: Not sure where these “673 phages” are coming from?

This is the total number of crAss-like phage genomes analyzed in this work, both previously identified and cMAGs (Supplementary Table S1).

11. L105: What is a reliable spacer match?

90% of spacer length, 90% identity, as explained in Methods: “The results of the BLASTN search of 673 crAss family assemblies against the spacer databases were analyzed as follows: a link between a virus assembly and a contig (genome partition) containing a CRISPR array was established when BLASTN yielded one or more hits with at least 90% of the spacer positions matching the corresponding phage sequence or two or more hits with at least 80% identity. Then, the source organism of the contig with the highest-scoring hit was selected as the potential host, with the preference given to spacers derived from completely sequenced genomes.”

12. L106: Replace “the” with “than”.

Done

13. L107: I did not have access to Supplementary Tables S1 and S2, so I was unable to give comments. It would be of interest to the readers to write some of these bacterial phyla in the text and perhaps comments on the meaning of these potential additional hosts.

Supplementary Tables S1 and S2 are provided in the revised text. We mention Firmicutes and Proteobacteria as the other bacterial phyla that are identified as hosts of crAss-like phages, although we do not have much to say about them except that they also are gut bacteria.

14. L111: 596-strong? Meaning?

removed

15. L120: Reference for this phage? See Journal of Aquatic Animal Health 31:225–238, 2019. Flavobacterium-infecting phage Fpv3 seems to belong to the (current) Podoviridae family. crAss phages are apparently also podophages. As far as I know Flavobacterium do belong to the Bacteroidetes phylum. Are Flavobacterium found in the human gut? If yes... Still, I see the point of excluding this one (one out of 596). Would this suggest a food origin?

As we indicate, “this might not be a native human gut phage”. Yes, we suggest the food origin of the Flavobacterium phage-related cMAG . We added RefSeq ID of Flavobacterium phage Fpv3 to the text.

16. L120: Does Flavobacteroides exist?

Corrected

17. L129: Replace “EpsiIn” with “Epsilon”.

Corrected

18. L129-132: Considering that Cellulophaga and Methylophaga hosts are mostly found in marine/sea environments, does this suggest that these Epsilon crAss phages may not be native human gut phages?

19. L132-138: Obviously, such larger genome would have to fit in the phage capsid. Any significant difference in the morphogenesis capsid/head genes/proteins? With such larger genomes, one may argue/wonder if Epsilon and Zeta should still be considered crAss-like phages?

No immediate observations on differences in structural proteins. Exhaustive analysis of the protein sequences is beyond the scope of this work. Certainly, it is legitimate question, whether the phages in the epsilon and zeta groups are “crAss-like”. There is an inevitable degree of arbitrariness in taxonomic assignments. Nevertheless, these phages, despite the peculiarities of genome architectures and inferred expression strategies, clearly belong in a clade with alpha, beta, gamma, delta (see all the trees in the work) and have similar gene compositions including signature genes, such as genes 74 and 75 in the structural module (see Fig. 1C). Therefore, this entire group is expected to become a new higher rank taxon (and order, most likely), and we believe it is natural to call it ‘expanded crAss-like phage assemblage’ or simply, ‘crAss-like phages’, for brevity. This is repeatedly explained in the manuscript.

20. L139-141: Would this be expected considering that most phages have similar discernible block of genes and morphogenesis genes are often conserved in related phages.

The presence of these gene blocks is expected, indeed, and no claim to the contrary is being made. This clause opens a paragraph that itemizes genes within these conserved blocks – part of these are unique to crAss-like phages.

20. L151: Replace “all crAss-like genes” with “all crAss-like phages”.

Corrected

21. L170: seems to possess?

We modified the sentence in question to “By contrast, phages in the Epsilon group encode a minimal set of replication machinery components”, to emphasize that the statement is based on genome sequence analysis. In this context, ‘seems’ does not appear to be necessary.

22. L173: Epsilon.

Corrected

23. L178: Replace “phi 14:2 of confirms” with “phi 14:2 confirms”.

Corrected

24. L183: “many phages”. Do these include podophages?

We meant crAss-like phages; sentence corrected

25. L185: Indeed, the presence of such nuclease in the structural block seems odd. Is this common in bacterial viruses or is limited to a group of phages?

This not common among phages, we indicate that in the revised manuscript.

26. L190: One would assume that the targeting spacers were in arrays associated to specific types/subtypes of CRISPR-Cas systems?

We did not detect any specific trends, those spacers are mostly in type I CRISPR loci, the most common variety. This is of no direct relevance to the subject of the paper, so we decided not to include these data.

27. L193-195: Are there other phages for which unknown orfs were proven to target host defense systems?

Yes, indeed, anti-CRISPR proteins have been discovered in this very manner. We clarify that in the revision.

28. L208-239: While the case for two DNAPs has been elegantly demonstrated, it is still unclear why such switching/displacement occurred in the first place and its role in the biology of crAss-like phages?

We discuss this as best we can, proposing that the DNAP switch has to do with a host defense mechanism that targets the phage DNAP. As per the reviewer's suggestion, this hypothesis has been removed from the abstract as too speculative, but we keep it in the Discussion. Clearly, this is amenable to experimental testing.

29. L229-230: Replace "appears to have involved" with "appears to have been involved".

Corrected

30. L241: This is a really interesting section. Wish it could have been confirmed by some proteomic work. Obviously can't for now.

We agree. Hopefully, our work stimulates other labs to undertake such analyses.

31. L250/L254: Some? How many? Are the numbers in Fig. 1B? Seems really unique to Delta and Zeta groups.

This is extremely difficult to quantify because we do not have formal criteria for "regions that do not encode any recognizable proteins but rather are occupied by short ORFs of different polarities", and we do not see a straightforward way to develop one. We notice the phenomenon and show some examples that are more or less clearly visible (see genome maps in Figure 2).

32. L263-265: Any example with characterized phages (instead of only metagenomic analyses)?

To the best of our knowledge, this has not been studied with characterized phages in any detail.

33. L270: Beta? They don't seem to have suppressor tRNA (Fig. 1B).

Figure 1B identifies one beta group genome with suppressor tRNA; moreover, absence of suppressor tRNA does not necessarily mean that the genome cannot use an alternative code as we point out in the paper.

34. L273: late genes as late-expressed genes? I would assume these are putative late-expressed genes? Any reason why this would be only for structural proteins? Presumably because it needs time to produce those suppressor tRNAs and translate a significant amount of structural proteins?

Corrected to "putative late genes" out of caution; indeed, these are late-expressed genes but 'late' is a common and clear term in virus research. The primary reason is that the early genes must be expressed using either host machinery or virion components. Because suppressor tRNAs are not known to accumulate in virus particles, they have to be expressed early in infection so that alternative codon assignment can be used only in late genes. It is a possibility, of course, that alternative coding disrupts the translation of host mRNAs while allowing the late phage mRNAs to be faithfully translated.

35. L280: prefect? perfect?

Corrected

36. L284: Replace "emerges in one a phage genome" with "emerges in a phage genome".

Corrected

37. L295: Any hypothesis on why such alternative coding path?

Indeed, we propose one in the Discussion: “Recoding enabled by the capture and adaptation of the suppressor tRNA can be perceived as an anti-defense strategy to impair the production of host proteins, including those involved in defense, by stop-codon readthrough, while faithfully translating the phage proteins.” We are inclined to think that elaborating further would be considered unwarranted speculation.

38. L297/309: Infestation? Consider revising.

We do not see what is wrong with the word. These are parasitic elements, so they infest the host genomes. It seems to be clear.

39. L299: Group II introns have been found in phages? Reference would be needed for this.

Yes, they are present in the genomes of some phages as mentioned in the first sentence of the respective section. We added two extra references.

40. L307: Replace “Zzeta” with “Zeta”.

Corrected

41. L310: Not only does it severely complicate annotation but I also wonder how it would impact their fitness? Any other characterized phages with such relatively high number of introns, either in the mcp or terL or rnap?

Given how abundant and apparently thriving the crAss-like phages are, this feature doesn’t seem to have any adverse effect on their fitness (the opposite, if anything). We are clearly saying that this is “unprecedented”; we are unaware of any comparable examples. This one of the foremost observations in this work as indicated in the abstract.

42. L311: Define MCP and it should be mcp in italic.

We defined MCP at the first appearance (“Conserved and group-specific genomic features in the extended crAss-like phage assemblage” section). We are using “MCP” as an acronym rather than a gene name, so it stays non-italic capitalized.

43. L333: I did not find any elements in the Results section regarding the “12% of the diversity” mentioned in the Discussion.

To clarify we added “596 of the 4907 distinct cMAGs” which is 12.2%.

44. L355-358: Are there proofs of such anti-defense strategy in the literature?

No proof which is why we are clearly indicate that this is our conjecture.

45. L384: Remove “.” after ref 11, replace “workand” with “work and”

Corrected

46. L390: Remove “.” after ref 11.

Corrected

47. L425: Replace “database the consisted” with “database that consisted”. Also in your Results sections, there is no mention of bacterial and archaeal abundance, however your databased included bacterial and archaeal genomes.

The typo is corrected. Bacterial and archaeal abundances were not estimated.

48. L346: Caudovirecetes? Reference? Google and PubMed searches revealed no hit (except your bioRxiv manuscript)?

De facto accepted: <https://www.ncbi.nlm.nih.gov/Taxonomy/Browser/wwwtax.cgi?id=2731619>

49. L345-348: If they all become families, one makes you wonder if they will still be called like crAss-like phages?

ICTV will decide the ultimate nomenclature and “crAss-like phages” probably won’t be a part of it. There is no reason to doubt, however, that it would still be perfectly fine to call them such as a colloquial term. There is a good chance that an order Crassvirales will be created.

50. L351-352. As written, this seems to reduce the impact of the previous work? Consider revising or remove.

We cannot at all see how the statement on simple genome organization and conventional genome strategy reduces the impact of the previous work (that of Dutilh et al 2014 or Yutin et al 2018). We think that it is a rather important point to make on the strategies that are to be used for the discovery and characterization of novel phages. Therefore, in this case, we do not see any reason to either modify or drop the original text.

51. L356-358: Not very convincing. Defense strategies would already be in place when phage infects the cell. However, it could be used to redirect the host machinery to produce phage proteins.

We must respectfully disagree. Defense strategies are well known to be activated in the course of phage infection (CRISPR is a good example. While the idea that using an alternative code redirects the translation system seems intuitively plausible, it is not actually clear how that would work mechanistically. On the contrary, the proposed mechanisms of impairing the functionality of host proteins is mechanistically straightforward.

52. L359: Still not sure how such a massively intron/intein-containing phage would be functional or efficiently replicate (fitness costs, etc).

We observe this genome organization, and then, there is the clear evidence of the diversity and high abundance of the Zeta group crAss-like phages, suggesting they are more than capable so surmount the problems. We can offer no convincing hypothesis on the mechanisms of their gene expression at this point. We agree, however, that this was not sufficiently emphasized in the original manuscript.

Therefore, we adding the following in the Discussion: “Clearly, given the high abundance of the Zeta group phages, they manage to replicate efficiently despite the genome infestation with mobile

elements. Experimental investigation of these phages has the potential to reveal novel mechanisms of gene expression.”

53. L365: Could this be due to the host they are infecting and not related to a defense mechanism?

Given that phages with different DNAPs seem to infect a well intermixed range of hosts, we consider it highly unlikely that these hosts have conflicting preferences for PolA over PolB, or vice versa. The distribution of host defense systems, to the contrary, is very idiosyncratic, so we find our hypothesis more plausible.

54. References: Are bioRxiv papers accepted (see ref 29)?

Not as of November 2nd, 2020

55. Figure 1: Replace “shaded” with “highlighted”.

Corrected

56. Figures 1 and 2 are well-designed and easy to understand.

We appreciate the comment.

57. Figure 3: Why did you decide to combine results for the transcription gene block and for the stop codons in the same figure? I don’t get the main message of this Figure at the first glance. I think it would be best to remove the stop codon locations from this figure, since it is represented in Figure 6.

To show how alternative coding, introns, and inteins complicate the annotation of this part of the genome

58. The authors should consider adding Supp Fig. 7 to the manuscript.

We find it best to keep it in the Supplement to keep the paper concise

REVIEWERS' COMMENTS

Reviewer #1 (Remarks to the Author):

The authors provided a convincing and extensive addressing of all our revision points. We are happy with those and we are satisfied with the current shape of the paper. There are still a few points we think should be addressed or better explained, but these are minor points.

Revision points

L88: cMAGs are likely to contain viruses and plasmids, but there is no guarantee that they mostly contain such elements. Authors should explain why they expect this. Also in light of the comment 3) of Rev#3, this may well be an element of general knowledge for some, but it should be better clarified. Is there any reference to this?

L299 OR L161: by looking at the trees reconstructed for different markers (gene75, MCP, portal, TerL), we noted that the Zeta group systematically had long branches. We wonder if this could be related to the particularities in the genetic code and/or the genome structure and length that you found for this group, or if this could be related to other issues. Therefore, we suggest to add something to further explain this pattern, as this could be an artifact due to saturation or other reasons (e.g., bias in the base composition, heterogeneity among branches); it may also be that there's a biological meaning for this observed difference.

L223 and L452: In the mapping of crAssphages against SRA data, authors use the term "gut metagenomes". However, in the accession numbers of Table S2, there are also enriched viromes (e.g. SRR828661, SRR935341). Please specify this in the methods and discussion, or remove the VLP viromes from the analyzed SRA metagenomes.

Reviewer #2 (Remarks to the Author):

The authors adequately addressed my concerns #1 and #3. Some questions remain relating to my concern #2. Specifically:

- It's unclear what the authors mean in their response by "Rarefaction analysis (random independent subsampling of genomes) works only under the assumption that the tree of the analyzed genomes is a star."

- I understand that new assembly methods and new sequencing projects are a driver of the increasing number of identified crAss-like genomes, but given the 100s of genomes that have been identified to date, some metric of saturation must be able to be computed (even if rarefaction analysis isn't the appropriate method). I would argue that this is not beyond the scope of the study, as a major takeaway of the work is that there are a lot of phages and that they are diverse. Notably, some flavor of this statement is present in the introduction of nearly every phage paper published in the past decade. So, some level of refinement of this conclusion is merited. Does this paper finally reveal the extent of interesting biology that crAss-like phages participate in? Or is there more biology likely to be discovered? Should efforts continue into sequence-based analyses of these phages or should efforts be diverted to experimental approaches now that we have n% of the crAss-like phages of the world sequenced?

- If the issue is that crAss-like phages are too diverse to make meaningful calculations, can analyses be conducted on sub-families? Are some of the sub-families truly larger than others or are they just more heavily sampled?

As such, at this stage, this is a stale conclusion that merits further calibration in order to be interesting and impactful. One way the authors can do this (without the need for new experiments) would be to predict whether continued exploration of crAss-like genomes is likely to yield more biology or if we're close to a saturated sample of crAss-like phages.

Specific points:

1) In my previous comment to the authors, I suggested rarefaction analysis. I feel that this was unfairly dismissed as misleading. Why would this be misleading?

2) An alternative approach could be to illustrate how many new genes are identified with each new crAss-like phage (see Hatfull et al. PLoS Genetics 2006, Figure 6).

3) The authors mentioned another method of assaying gene flux, but said it was beyond the scope of the study. I'm not familiar with this analysis... Why is it beyond the scope of the study?

At a minimum, it should be important that point #2, above, is addressed. If this is not possible, the authors should provide a more clear explanation of the caveats of why these analyses cannot be done (with data) and a better discussion highlighting the novel findings -beyond the catalog of genomic features- in the context of previous work, some provided below.

- Yutin et al Nature Microbiology 2017
- Guerin et al Cell Host & Microbe 2018
- Edwards et al Nature Microbiology 2019
- Hryckowian et al Cell Host & Microbe 2020

Reviewer #3 (Remarks to the Author):

Thank you for the modifications to the manuscript.

Minor comments

Check References/Titles of publications: Only the first word of the title should be capitalized.

Page numbers are missing in a few references.

Reviewer #1 (Remarks to the Author):

The authors provided a convincing and extensive addressing of all our revision points. We are happy with those and we are satisfied with the current shape of the paper. There are still a few points we think should be addressed or better explained, but these are minor points.

Revision points

L88: cMAGs are likely to contain viruses and plasmids, but there is no guarantee that they mostly contain such elements. Authors should explain why they expect this. Also in light of the comment 3) of Rev#3, this may well be an element of general knowledge for some, but it should be better clarified. Is there any reference to this?

Response: *The circularity of the cMAGs ensures enrichment for virus and plasmid sequences. This is indeed general understanding that we additionally emphasized in the current revision. Search for virus marker genes allows us to identify the virus subset of cMAGs.*

L299 OR L161: by looking at the trees reconstructed for different markers (gene75, MCP, portal, TerL), we noted that the Zeta group systematically had long branches. We wonder if this could be related to the particularities in the genetic code and/or the genome structure and length that you found for this group, or if this could be related to other issues. Therefore, we suggest to add something to further explain this pattern, as this could be an artifact due to saturation or other reasons (e.g., bias in the base composition, heterogeneity among branches); it may also be that there's a biological meaning for this observed difference.

Response: We agree with the reviewer that the long branches in the Zeta group merits some attention. In the revised Discussion, we write: "In the phylogenetic trees of conserved phage proteins, the Zeta group usually forms long branches (Figure 1A and Suppl figures 3 and 4), suggesting that evolution of the unusual features of the genome architecture of these phages is accompanied by fast sequence evolution. Such apparent coupling between sequence and genome organization evolution rates seems to be best compatible with weak selection that makes these phages vulnerable to MGE."

L223 and L452: In the mapping of crAssphages against SRA data, authors use the term "gut metagenomes". However, in the accession numbers of Table S2, there are also enriched viromes (e.g. SRR828661, SRR935341). Please specify this in the methods and discussion, or remove the VLP viromes from the analyzed SRA metagenomes.

Response: *We appreciate this point and write of "metagenomes and enriched viromes" in the revision.*

Reviewer #2 (Remarks to the Author):

The authors adequately addressed my concerns #1 and #3. Some questions remain relating to my concern #2. Specifically:

- It's unclear what the authors mean in their response by "Rarefaction analysis (random independent subsampling of genomes) works only under the assumption that the tree of the analyzed genomes is a star."

- I understand that new assembly methods and new sequencing projects are a driver of the increasing number of identified crAss-like genomes, but given the 100s of genomes that have been identified to date, some metric of saturation must be able to be computed (even if rarefaction analysis isn't the appropriate method). I would argue that this is not beyond the scope of the study, as a major takeaway of the work is that there are a lot of phages and that they are diverse. Notably, some flavor of this statement is present in the introduction of nearly every phage paper published in the past decade. So, some level of refinement of this conclusion is merited. Does this paper finally reveal the extent of interesting biology that crAss-like phages participate in? Or is there more biology likely to be discovered? Should efforts continue into sequence-based analyses of these phages or should efforts be diverted to experimental approaches now that we have n% of the crAss-like phages of the world sequenced?

- If the issue is that crAss-like phages are too diverse to make meaningful calculations, can analyses be conducted on sub-families? Are some of the sub-families truly larger than others or are they just more heavily sampled?

As such, at this stage, this is a stale conclusion that merits further calibration in order to be interesting and impactful. One way the authors can do this (without the need for new experiments) would be to predict whether continued exploration of crAss-like genomes is likely to yield more biology or if we're close to a saturated sample of crAss-like phages.

Specific points:

- 1) In my previous comment to the authors, I suggested rarefaction analysis. I feel that this was unfairly dismissed as misleading. Why would this be misleading?
- 2) An alternative approach could be to illustrate how many new genes are identified with each new crAss-like phage (see Hatfull et al. PLoS Genetics 2006, Figure 6).
- 3) The authors mentioned another method of assaying gene flux, but said it was beyond the scope of the study. I'm not familiar with this analysis... Why is it beyond the scope of the study?

At a minimum, it should be important that point #2, above, is addressed. If this is not possible, the authors should provide a more clear explanation of the caveats of why these analyses cannot be done (with data) and a better discussion highlighting the novel findings -beyond the catalog of genomic features- in the context of previous work, some provided below.

Response: Given the multiple pertinent questions asked by the reviewer regarding the projected diversity of the crAss-like phages, we performed the rarefaction analysis and included the results as the new Figure 4. In the revised Results, we write: "Gene complements of crAss-like phages are highly diverse both within and across groups such that about 20% of the genes in each phage genome have no detectable homologs in other members of the crAss-like assemblage. Rarefaction analysis indicates that the currently sampled crAss-like pangenome is far from saturation (Figure 4), so that numerous unique genes are expected to appear in additional crAss-like phage genomes." Predicting how many new families and how much biological novelty remain to be discovered is much harder. In the revised Discussion, we indicate: "How many additional crAss-like phages and, perhaps more importantly, how many distinct groups of such phages can we expect to discover in the future? The 596 crAss-like phage genome assemblies neatly fall into five distinct groups, without many unassigned singletons, suggesting that, within the sampled metagenome diversity, the most common divisions of crAss-like phages are already known. Similarly, these 596 genomic assemblies form 221 clusters, roughly at the species level, which implies that a non-negligible fraction of crAss-like phage species has been identified. It remains to be seen how much more crAss-like phage diversity is brought about by the analysis of the vastly expanded collection of increasingly diverse metagenomes that can be anticipated to become available within the next few years." We believe that by making these additions to the paper we addressed the reviewers questions and comments to the maximum extent realistically possible.